# How Muon's Spectral Design Benefits Generalization: A Study on Imbalanced Data

**Bhavya Vasudeva**[🌴], **Puneesh Deora**[🌲], **Yize Zhao**[🌲], **Vatsal Sharan**[🌴], **Christos Thrampoulidis**[🌲]

[🌴]University of Southern California  [🌲]University of British Columbia
bvasudev@usc.edu, {puneeshdeora,cthrampo}@ece.ubc.ca

## Abstract

The growing adoption of spectrum-aware matrix-valued optimizers such as Muon and Shampoo in deep learning motivates a systematic study of their generalization properties and, in particular, when they might outperform competitive algorithms. We approach this question by introducing appropriate simplifying abstractions as follows: First, we use imbalanced data as a testbed. Second, we study the canonical form of such optimizers, which is Spectral Gradient Descent (SpecGD)—each update step is $UV^T$ where $U\Sigma V^T$ is the truncated SVD of the gradient. Third, within this framework we identify a canonical setting for which we precisely quantify when SpecGD outperforms vanilla Euclidean GD. For a Gaussian mixture data model and both linear and bilinear models, we show that unlike GD, which prioritizes learning dominant principal components of the data first, SpecGD learns all principal components of the data at equal rates. We demonstrate how this translates to a growing gap in class balanced loss favoring SpecGD early in training and further show that the gap remains consistent even when the GD counterpart uses adaptive step-sizes via normalization. By extending the analysis to deep linear models, we show that depth amplifies these effects. We empirically verify our theoretical findings on a variety of imbalanced datasets. Our experiments compare practical variants of spectral methods, like Muon and Shampoo, against their Euclidean counterparts and Adam. The results validate our findings that these spectral optimizers achieve superior generalization by promoting a more balanced learning of the data's underlying components.

## 1 Introduction

Spectrum-aware optimizers such as Shampoo (Gupta et al., 2018), Muon (Pethick et al., 2025; Jordan et al., 2024) have recently gained significant traction in the deep learning community, delivering substantial training speedups for deep classifiers (Jordan et al., 2024) and transformer language models (Vyas et al., 2025; Liu et al., 2025) compared to standard methods like SGD (Robbins & Monro, 1951) with momentum or even Adam (Kingma & Ba, 2014). The key distinction lies in how these methods treat neural network parameters: while SGD and Adam operate entry-wise on vectorized parameters, Shampoo and Muon work directly with matrix-valued parameters (e.g., weight matrices and attention matrices) at the layer level (Carlson et al., 2015; Large et al., 2024). This matrix-level approach intuitively enables optimization trajectories that entry-wise methods cannot achieve. Despite their empirical success, a fundamental question remains unanswered: **when do spectrum-aware optimizers generalize better than standard methods?**

The challenge is substantial. Even well-established optimizers like Adam, despite more than a decade of practical dominance, remain less well understood than SGD, whose Euclidean GD trajectory is well-characterized both statistically and algorithmically. Recent theoretical progress has begun to illuminate these methods through the lens of implicit bias. For instance, while the implicit bias of GD toward $\ell_2$-norm max-margin classifiers is well-established (Soudry et al., 2018; Ji & Telgarsky, 2018), recent work proved that Adam converges toward $\ell_\infty$-norm max-margin classifiers in linear settings (Zhang et al., 2024; Xie & Li, 2024). Conceptually, this difference can be understood by noting that Adam reduces to SignGD when momentum and preconditioning histories are ignored (*i.e.*, $\beta$ parameters set to zero). This reduction has often been leveraged before to study Adam's properties, since SignGD is simpler to analyze, e.g., Kunstner et al. (2024); Vasudeva et al. (2025b).

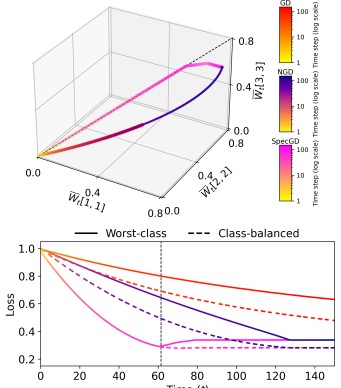

Figure 1: (Left) Test accuracy for training an MLP on Colored-MNIST with a $99\%$ digit–colour correlation using NMD, Signum, and Muon (see text for details). All optimizers achieve near-perfect accuracy on majority groups (same digit–colour labels), but Muon outperforms the others on minority groups (opposite digit–colour labels), especially early in training. (Middle) Eigenvalues and eigenvectors of the empirical moment matrices for all training images show that the dominant spectral components correspond to colour. (Right) Reconstructions of sample images reveal that digit-shape information lies in less dominant components. This suggests that Muon's spectral design promotes balanced learning of these components, leading to superior generalization.

A similar reduction exists for spectrum-aware optimizers: Shampoo and Muon without accumulations and with exact matrix operations reduce to Spectral Gradient Descent (SpecGD) (Pethick et al., 2025; Bernstein & Newhouse, 2024), where each update step is $UV^T$, with $U\Sigma V^T$ denoting the truncated SVD of the gradient (see Sec. 3 for details). Thus, just as SignGD serves as the canonical form for understanding Adam, SpecGD serves as a canonical form for understanding Shampoo and Muon. Moreover, both SignGD and SpecGD are instances of normalized steepest descent with respect to max and spectral norms, respectively (Bernstein, 2009; Carlson et al., 2015; Pethick et al., 2025).

Recently, Fan et al. (2025); Tsilivis et al. (2024) characterized SpecGD's optimization bias in linear multi-class and homogeneous binary classification, showing that it drives weights toward a max-margin classifier with respect to the spectral norm. However, these results have two limitations. First, they only describe the algorithm's behavior in the terminal phase of training, which may not reflect practical deep learning scenarios that often employ early stopping. Second, and more importantly, implicit bias results provide no direct guarantees about generalization performance—the ultimate objective in machine learning. For instance, even in linear settings, the minimum spectral norm solution may not be unique. Recent work has also studied Muon and Shampoo, but primarily from the perspective of optimization properties (Pethick et al., 2025; Morwani et al., 2024; Vyas et al., 2025; Li & Hong, 2025; Chang et al., 2025; Chen et al., 2025) or scalability (Liu et al., 2025; Boreiko et al., 2025), leaving their generalization behavior and underlying mechanisms comparatively underexplored.

Motivated by the apparent lack of understanding of the generalization properties of SpecGD, we ask: **Can we identify concrete settings where SpecGD generalizes better than standard (Euclidean) GD?**

**Imbalanced Data as Playground.** We introduce imbalanced data as a testbed for studying SpecGD's potential generalization advantages. Fig. 1 provides a concrete

Figure 2: Dynamics of the iterates (top) and loss (bottom) for GD, NGD, and SpecGD on a linear model with class imbalance (priors $p_1 > p_2 > p_3$). SpecGD learns all spectral components at the same rate, whereas (N)GD prioritizes more dominant components. Although all converge to the same solution, SpecGD significantly outperforms in worst-class and class-balanced loss early-on in training. See Sec. 3.2 for a precise characterization of SpecGD dynamics and a proof of its superiority over (N)GD.

demonstration: we train an MLP on Colored-MNIST (Arjovsky et al., 2020; Pezeshki et al., 2021) under severe group imbalance, where each digit appears in its majority color $99\%$ of the time during training, creating a strong spurious digit-color correlation (see App. D.1 for details). At test time, we evaluate on majority and minority groups with same and opposite digit-color associations, respectively. The results are revealing: whereas Muon improves on both groups early in training, its Frobenius and max norm counterparts, namely normalized momentum descent (NMD) and Signum (Bernstein et al., 2018), respectively, take much longer to improve on the minority group. We observe

a similar effect on the CIFAR dataset (Krizhevsky et al., 2009) with class imbalance (see Sec. 2 for details).

**Theoretical Comparison in Class-imbalanced Data.** To demystify this behavior and analyze the implicit regularization of different update rules, we analyze in detail a linear model trained with squared-loss under class imbalance. We derive closed-form expressions for the training trajectories of (Euclidean) GD and SpecGD, and use them to show that, with early stopping, SpecGD achieves a lower worst-class and class-balanced risk than GD, whereas both approach the same risk asymptotically in time. We also show that introducing adaptive step-size in GD via normalization does not equalize the gap: SpecGD outperforms normalized GD (NGD) as well. This is illustrated in Fig. 2.

Extending to deep networks, we derive the closed-form training trajectory for SpecGD with $L \geq 1$ layers. Our findings reveal two key effects of depth: i) it accelerates the learning of all spectral components, and ii) it narrows the time gap between the saturation points of different components.

**Experimental Results.** We empirically validate our theoretical insights through experiments on datasets with class or group imbalance. We demonstrate that practical variants of SpecGD, such as Muon and Shampoo, achieve superior generalization over SGD by promoting a more balanced learning of the spectral components of the data. While our theory focuses on uncovering the distinct learning mechanisms of SpecGD and GD, our experiments also include Adam to provide a comprehensive benchmark against a commonly-used baseline.

## 2 RESULTS ON CIFAR WITH STEP-IMBALANCE

In this section, we conduct a preliminary experiment to investigate whether Muon's superiority observed in the group-imbalanced setting in Fig. 1 also persists under class imbalance. We consider step-imbalanced CIFAR-10 and CIFAR-100 (Krizhevsky et al., 2009) with imbalance ratio 20 (majorities contain $20\times$ more samples than minorities). We train ResNet-18/50 on CIFAR-10/100 with three optimizers: NMD, Signum, and Muon (see App. D.2 for details and additional results with lower imbalance ratios). As shown in Fig. 3, Muon achieves superior minority-class performance compared to NMD and Signum. While all optimizers achieve similar majority-class accuracy, Muon more effectively reduces the performance gap between minority and majority classes, particularly early on in training. This result further suggests that spectral methods may offer advantages in imbalanced settings, motivating our subsequent theoretical investigation into the mechanisms underlying this behavior.

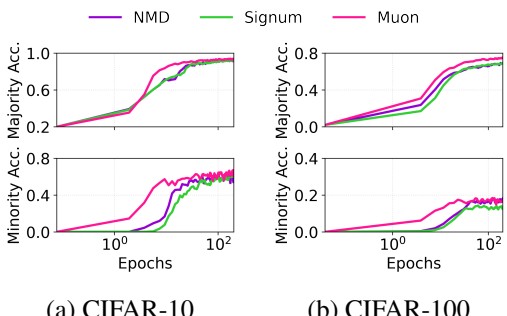

(a) CIFAR-10       (b) CIFAR-100

Figure 3: Test accuracy dynamics for training ResNet-based models on (a) CIFAR-10, and (b) CIFAR-100, both with STEP imbalance (20:1 majority-to-minority class ratio) using NMD, Signum, and Muon. All optimizers achieve comparable accuracy on majority classes, but Muon outperforms the others on minority-class performance, especially early in training.

## 3 LINEAR MODEL ON CLASS-IMBALANCED DATA

**Notation.** We denote matrices, vectors and scalars by $\boldsymbol{A}$, $\boldsymbol{a}$, and $a$, respectively. We denote the $(i,j)$-th entry of matrix $\boldsymbol{A}$ as $A[i,j]$. Let $\|\cdot\|_F$, $\|\cdot\|_{\max}$, and $\|\cdot\|_2$ denote the Frobenius, max and spectral norms, respectively, where $\|\boldsymbol{A}\|_{\max} := \max_{i,j} |A[i,j]|$. Let $\|\boldsymbol{a}\|_2$ denote the $\ell_2$ norm of $\boldsymbol{a}$. $\mathbb{1}\left[\cdot\right]$ denotes the indicator function, *e.g.*, $\mathbb{1}\left[a \geq b\right] = 1$ if $a \geq b$ and 0 otherwise. For matrices $\boldsymbol{A}, \boldsymbol{B}$, the inner product $\langle \boldsymbol{A}, \boldsymbol{B} \rangle = \mathrm{tr}(\boldsymbol{A}\boldsymbol{B}^\top)$.

Let $\boldsymbol{W} \in \mathbb{R}^{K \times d}$ denote the weight matrix of a linear model, and $\mathcal{L}(\boldsymbol{W})$ denote the loss. Let $\boldsymbol{W}_t$ and $\boldsymbol{\nabla}_t := \nabla\mathcal{L}(\boldsymbol{W}_t)$ denote the iterate and gradient at time $t$, respectively. Normalized steepest-descent (NSD) updates with respect to norm $\|\cdot\|$, with step-size $\eta > 0$, are (Boyd & Vandenberghe, 2009):

$$\boldsymbol{W}_{t+1} = \boldsymbol{W}_t - \eta\boldsymbol{\Delta}_t, \text{ where } \boldsymbol{\Delta}_t := \mathrm{argmax}_{\|\boldsymbol{\Delta}\| \leq 1} \langle \boldsymbol{\nabla}_t, \boldsymbol{\Delta} \rangle. \tag{1}$$

As discussed in Sec. 1, NGD, SignGD and SpecGD are instances of normalized steepest descent Eq. (1) with respect to Frobenius, max and spectral norm, respectively. Normalized momentum steepest-descent (NMD) updates are defined similarly as Eq. (1) but use momentum $\boldsymbol{M}_t = \beta\boldsymbol{M}_{t-1} +$

Table 1: Normalized steepest descent (NSD) and normalized momentum descent (NMD) updates for different norms. Here, $\boldsymbol{\nabla}_t$ denotes the gradient, $\boldsymbol{M}_t = \beta \boldsymbol{M}_{t-1} + (1-\beta)\boldsymbol{\nabla}_t$ denotes the momentum term, and $\boldsymbol{\nabla}_t = \boldsymbol{U}_t \boldsymbol{\Sigma}_t \boldsymbol{V}^\top$ and $\boldsymbol{M}_t = \widetilde{\boldsymbol{U}}_t \widetilde{\boldsymbol{\Sigma}}_t \widetilde{\boldsymbol{V}}^\top$ denote their SVDs. Setting $\beta = 0$ in the NMD updates yields the corresponding NSD updates.

| Norm | NSD update $\boldsymbol{\Delta}_t := \operatorname{argmax}_{\|\boldsymbol{\Delta}\| \leq 1} \langle \boldsymbol{\nabla}_t, \boldsymbol{\Delta} \rangle$ | NMD update $\boldsymbol{\Delta}_t := \operatorname{argmax}_{\|\boldsymbol{\Delta}\| \leq 1} \langle \boldsymbol{M}_t, \boldsymbol{\Delta} \rangle$ |
|---|---|---|
| $\|\cdot\|_F$ | NGD: $\boldsymbol{\Delta}_t = \dfrac{\boldsymbol{\nabla}_t}{\|\boldsymbol{\nabla}_t\|_F}$ | NMD: $\boldsymbol{\Delta}_t = \dfrac{\boldsymbol{M}_t}{\|\boldsymbol{M}_t\|_F}$ |
| $\|\cdot\|_{\max}$ | SignGD: $\boldsymbol{\Delta}_t = \texttt{sign}(\boldsymbol{\nabla}_t)$ | Signum: $\boldsymbol{\Delta}_t = \texttt{sign}(\boldsymbol{M}_t)$ |
| $\|\cdot\|_2$ | SpecGD: $\boldsymbol{\Delta}_t = \boldsymbol{U}_t \boldsymbol{V}_t^\top$ | Muon: $\boldsymbol{\Delta}_t = \widetilde{\boldsymbol{U}}_t \widetilde{\boldsymbol{V}}_t^\top$ |

$(1-\beta)\boldsymbol{\nabla}_t$ in place of $\boldsymbol{\nabla}_t$. Table 1 summarizes the updates for NSD and NMD for the three norms. Note that Muon with exact matrix operations is NMD with spectral norm, and reduces to SpecGD when $\beta = 0$. Also see App. A, where we discuss how Shampoo and Adam without accumulations reduce to SpecGD and SignGD, respectively.

**Data Model.** Let $y \in \{1, \ldots, k\}$ denote the class labels, and let the corresponding one-hot labels be denoted as $\boldsymbol{y} \in \{\boldsymbol{e}_c\}_{c=1}^k$, where $\boldsymbol{e}_c$ is the $c$-th standard basis vector in $\mathbb{R}^k$. Class probabilities are given by $p_c := \Pr(y = c)$ such that $\sum_{c=1}^k p_c = 1$. Each class $c$ has an associated mean vector $\boldsymbol{\mu}_c \in \mathbb{R}^d$, and samples for class $c$ are generated as isotropic Gaussians with mean $\boldsymbol{\mu}_c$. Finally, we assume that the means $\boldsymbol{\mu}_c$ are orthogonal. Put together, the data model we study is such that:

$$\Pr(y = c) = p_c, \; c \in [k], \quad \boldsymbol{x}|y \sim \mathcal{N}\big(\boldsymbol{\mu}_y, \sigma_x^2 \mathbb{I}_d\big), \quad \text{and} \quad \boldsymbol{\mu}_1 \perp \ldots \perp \boldsymbol{\mu}_k, \|\boldsymbol{\mu}_c\| = \mu. \quad \text{(DM)}$$

Define minority class index $m := \arg\min_{c \in [k]} p_c$ with class prior $p_m$, and majority class index $M := \arg\max_{c \in [k]} p_c$ with class prior $p_M$. Let $\texttt{SNR} := \frac{\mu^2}{\sigma_x^2}$ denote the signal-to-noise ratio (SNR).

## 3.1 EXPERIMENTAL RESULTS

For the experiments, we consider a heavy-tailed class-imbalanced setting by choosing $p_c \propto \frac{1}{c}$, and we sample each $\boldsymbol{\mu}_c$ independently from a zero-mean isotropic Gaussian distribution and normalize it. We use 20 classes, 100 samples, $d = 200$ and $\sigma_x = 0.1$. We initialize the weight matrix $\boldsymbol{W}_0$ by sampling each entry indepen-

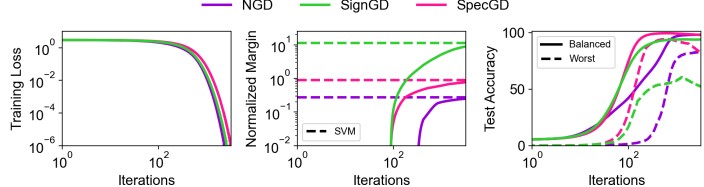

Figure 4: Results for a linear model trained with NGD, SignGD and SpecGD on heavy-tailed class-imbalance data using cross-entropy loss. Early-stopped SpecGD attains higher class-balanced and worst-class accuracy compared to other update rules or stopping points.

dently from $\mathcal{N}(0, \frac{1}{d})$. We use learning rates 0.025, 0.005 and $5 \times 10^{-4}$ for NGD, SignGD and SpecGD, respectively. These choices are made such that train loss curves of the algorithms are comparable. The training is stopped when the norm of the gradient for SpecGD falls below $10^{-6}$, as small gradients can introduce numerical imprecision in the SVD computation.

Fig. 4 shows results of training a linear model in this setting using NGD/SignGD/SpecGD to minimize cross-entropy loss. We observe that for all three update rules, as train loss approaches 0, the iterates $\boldsymbol{W}_t$ converge to a solution that maximizes the margin defined with respect to the corresponding norm (Fan et al., 2025). However, comparing test performance, early-stopped SpecGD attains higher class-balanced and worst-class test accuracies compared to NGD or SignGD at any stopping point.

## 3.2 THEORETICAL ANALYSIS

In this section, we analyze and compare the dynamics of GD and SpecGD. For tractability, we consider squared-loss, and population setting. Specifically, let $\mathcal{L}(\boldsymbol{W}) = \frac{1}{2}\mathbb{E}\|\boldsymbol{y} - \boldsymbol{W}\boldsymbol{x}\|_2^2$, where the expectation is over the joint distribution of $\boldsymbol{x}, \boldsymbol{y}$ in (say) (DM). In addition, define $\mathcal{L}_c(\boldsymbol{W}) = \frac{1}{2}\mathbb{E}_{\boldsymbol{x}|y=c}\|\boldsymbol{y} - \boldsymbol{W}\boldsymbol{x}\|_2^2$ to be the class-conditional loss for class $c \in [k]$ and let $\mathcal{L}_{\text{bal}}(\boldsymbol{W}) = \frac{1}{k}\sum_{c \in [k]} \mathcal{L}_c(\boldsymbol{W})$ be the balanced

loss. Define the population moment matrices $\boldsymbol{\Sigma_{xx}} := \mathbb{E}[\boldsymbol{x}\boldsymbol{x}^\top]$ and $\boldsymbol{\Sigma_{yx}} := \mathbb{E}[\boldsymbol{y}\,\boldsymbol{x}^\top]$. Our analysis holds when the (full) SVDs of these moment matrices are jointly diagonalizable.

**Condition 1** (Joint Diagonalizability). *There exist orthonormal matrices $\boldsymbol{U} \in \mathbb{R}^{k \times k}, \boldsymbol{V} \in \mathbb{R}^{d \times d}$, and matrices $\boldsymbol{S_{yx}} \in \mathbb{R}^{k \times d}, \boldsymbol{S_{xx}} \in \mathbb{R}^{d \times d}$ with non-zero entries only along their main diagonals* [*] $(s_1^{yx} \geq s_2^{yx} \cdots \geq s_k^{yx} \geq 0$ and $s_1^{xx} \geq s_2^{xx} \cdots \geq s_d^{xx} \geq 0$, respectively), such that, $\boldsymbol{\Sigma_{yx}} = \boldsymbol{U}\boldsymbol{S_{yx}}\boldsymbol{V}^\top$ and $\boldsymbol{\Sigma_{xx}} = \boldsymbol{V}\boldsymbol{S_{xx}}\boldsymbol{V}^\top$.*

Under this condition applied to the empirical moment matrices, Saxe et al. (2013); Gidel et al. (2019) derive closed-form training dynamics of two-layer linear networks. Here, we instead apply this condition on population statistics making it possible to study test statistics. In the lemma below, we show that our data model (DM) satisfies this condition. See App. B.1 for the proof.

**Lemma 3.1.** *The population moment matrices of data model (DM) satisfy Condition 1, with $s_c^{yx} = \mu p_c$ for $c \in [k]$, $s_c^{xx} = \mu^2 p_c + \sigma_x^2$ for $1 \leq c \leq k$, and $s_c^{xx} = \sigma_x^2$ for $k < c \leq d$.*

This result shows that the data model (DM) is one such setting where Condition 1 strictly holds. Notably, in Gidel et al. (2019), the authors validate whether a weaker version of this condition is satisfied for some datasets used in practice. Specifically, they check whether there exist orthonormal matrices $\boldsymbol{U}, \boldsymbol{V}$ such that the empirical moment matrices are jointly diagonalizable as $\boldsymbol{U}\boldsymbol{\Sigma_{yx}}\boldsymbol{V}^\top$ and $\boldsymbol{V}(\boldsymbol{\Sigma_{xx}} + \boldsymbol{B})\boldsymbol{V}^\top$, respectively, where $\|\boldsymbol{B}\|$ is much smaller than $\|\boldsymbol{\Sigma_{xx}}\|$. They find that on datasets like MNIST, CIFAR, etc., the ratio is indeed small (see Table 1 in Gidel et al. (2019)), so this can be considered as a reasonable condition. We consider Condition 1, where $\|\boldsymbol{B}\| = 0$, mainly to make the analysis more tractable. See App. C for further discussion, and experimental results that examine the effect of relaxing this condition.

**Evolution of $\boldsymbol{W}_t$.** We compare the evolution of the weight matrix $\boldsymbol{W}_t$ for GD and SpecGD over iterations $t$. For each iteration $t$, define $\overline{\boldsymbol{W}}_t := \boldsymbol{U}^\top \boldsymbol{W}_t \boldsymbol{V}$ and recall that $\overline{W}_t[c, c]$ denotes the $c^{\text{th}}$ main-diagonal entry of $\overline{\boldsymbol{W}}_t$, for $c \in [k]$.

Under Condition 1, Saxe et al. (2013) (see also App. B.2) shows that when initialized at zero and run with sufficiently small step size $\eta < \min_{c \in [k]}(s_c^{xx})^{-1}$, GD iterates $\boldsymbol{W}_t$ are such that $\overline{\boldsymbol{W}}_t$ is diagonal at each iteration with diagonal entries evolving as (and approximation becoming accurate for gradient flow (GF) $\eta \to 0$):

$$\overline{W}_t[c, c] = s_c^{yx} \frac{1 - (1 - \eta s_c^{xx})^t}{s_c^{xx}} \approx \frac{s_c^{yx}}{s_c^{xx}} \left(1 - e^{-\eta s_c^{xx} t}\right). \tag{2}$$

The following result characterizes the dynamics of $\boldsymbol{W}_t$ for SpecGD. See App. B.2 for the proof.

**Proposition 1.** *Assume zero initialization $\boldsymbol{W_0} = \boldsymbol{0}$ and Condition 1 holds. Then, for SpecGD with step size $\eta < \min_{c \in [k]} \frac{s_c^{yx}}{s_c^{xx}}$, at each iteration, $\boldsymbol{W}_t = \boldsymbol{U}\overline{\boldsymbol{W}}_t\boldsymbol{V}^\top$ where $\overline{\boldsymbol{W}}_t$ is zero except its main diagonal along which the entries evolve as follows for $c \in [k]$:*

$$\overline{W}_t[c, c] = \eta\, t\, \mathbb{1}\left[t \leq \frac{s_c^{yx}}{\eta s_c^{xx}}\right] + \frac{s_c^{yx}}{s_c^{xx}} \mathbb{1}\left[t > \frac{s_c^{yx}}{\eta s_c^{xx}}\right].$$

Comparing the above two displays, which contrast GD's and SpecGD's iterate evolution, shows the following. Although both methods asymptotically converge to the same solution, their trajectories differ significantly. GD learns component $c$ at a rate proportional to $s_c^{xx}$, meaning more dominant spectral components are learned faster. In contrast, SpecGD learns all components at the same rate until each individual value saturates and converges to its terminal value. This difference is also demonstrated in Fig. 2 (top), which tracks $\overline{W}_t[c, c]$ for GD and SpecGD in a class-imbalanced setting (data model (DM)) with $\eta = 0.01$, $d = k = 3$, $\mu = 1$, $\sigma_x^2 = 0.125$, and $p_1, p_2, p_3$ set as $0.5, 0.3, 0.2$, respectively.

In App. C, we compare the dynamics of GD and SpecGD derived under Condition 1 to a more general setting using the same data model but with finite samples and random initialization, and show that the theoretically derived dynamics closely match those observed empirically (see Fig. 11). Additionally, in App. C, we also compare the dynamics of Muon (with $\beta = 0.9$) to those of SpecGD, both when Condition 1 holds (see Fig. 10) and in the finite sample setting (see Fig. 12). In both cases, we find that similar to SpecGD, Muon promotes learning different spectral components at a similar rate.

---

[*]With slight abuse of terminology, we will refer to rectangular matrices having non-zero entries only on their main diagonals simply as 'diagonal' hereon.

**Generalization of SpecGD vs. (N)GD.** We now show that this property of SpecGD to learn concepts at equal rate translates to superior generalization compared to GD in an imbalanced setting where least-significant spectral components of the moment matrices are associated to minority classes. Concretely, under data model (DM), the $k$ first eigenvectors of $\Sigma_{xx}$ align with the class-mean vectors, ordered in decreasing class prior probability (see proof of Lemma 3.1). Intuitively, then, learning spectral components earlier during training should translate to generalization gains. Indeed, in Fig. 2 (bottom), we observe that SpecGD attains lower minority-class and class-balanced loss compared to GD. The following theorem formalizes this intuition. We use continuous-time versions of GD and SpecGD here, *i.e.*, gradient flow (GF) and spectral GF (SpecGF). The dynamics for GF are characterized by the approximation in Eq. (2), while the SpecGF dynamics follow Proposition 1, both with $\eta = 1$ (see App. B.2 for details). We use $W(t)$ and $\overline{W}(t)$ to denote the continuous-time iterates and the corresponding singular value matrix, respectively.

**Theorem 1** (Simplified). *Assume data model* (DM) *and zero initialization. Let $t^\star = \frac{s_m^{yx}}{s_m^{xx}}$ be the first time SpecGF fits the minority class $m$. Further, assume that $\mu \geq 1, k \geq 3\mu$, and $p_m \leq \frac{1}{5\,SNR+6k}$. Then, for every $t \in (0, t^\star]$, SpecGF outperforms GF with minority-class and balanced loss gaps between the algorithms growing linearly with time as follows:*

$$\mathcal{L}_m^{\text{GF}}(t) - \mathcal{L}_m^{\text{Spec}}(t) \geq \mu t/4, \quad \mathcal{L}_{\text{bal}}^{\text{GF}}(t) - \mathcal{L}_{\text{bal}}^{\text{Spec}}(t) \geq \mu t/2.$$

For clarity, we state the above theorem under some simplifications on the condition on minority class prior $p_m$. See App. B.3 for the complete version and the proof.

Next, we investigate whether these gains of SpecGD arise because it uses normalized updates while GD does not. To account for this difference, we compare SpecGD with NGD, the normalized version of GD. We first simulate the dynamics of NGD in the setting of Fig. 2, and find that although NGD converges much faster than GD, its iterates $\overline{W}_t$ follow the same trajectory as GD (Fig. 2 (top)). Note that at the end of training, all three optimizers eventually converge to the same solution. However, importantly, we find that early on in training, SpecGD still outperforms NGD in terms of both minority-class and class-balanced loss (Fig. 2 (bottom)). We formalize and prove this observation in the theorem below. See App. B.4 for the complete version and the proof.

**Theorem 2** (Simplified). *Assume data model* (DM)*, zero initialization, and gradient flow algorithms NGF and SpecGF. Further, assume that $p_m \leq \frac{1}{2\,SNR+4k}$, and $k \geq \frac{9}{p_M - p_m}$. Then, for every $t \in (0, t^*]$, SpecGF outperforms NGF with minority-class and balanced loss gaps between the algorithms growing linearly with time as follows:*

$$\mathcal{L}_m^{\text{NGF}}(t) - \mathcal{L}_m^{\text{Spec}}(t) \geq \mu t/2, \quad \mathcal{L}_{\text{bal}}^{\text{NGF}}(t) - \mathcal{L}_{\text{bal}}^{\text{Spec}}(t) \geq \mu t/2.$$

**Proof Sketch.** We now provide a brief proof sketch for Theorem 1 highlighting the key ideas. From Proposition 1, since every SpecGF iterate $W(t)$ remains in the same basis $U, V$, i.e., $W(t) = U\overline{W}(t)V$, the singular values in $\overline{W}(t)$ are the key quantities of interest for bounding the loss. Concretely, for any class $c \in [k]$, we can relate its loss to the $c^{\text{th}}$ singular value $\overline{W}(t)[c, c] =: \alpha_c(t)$:

$$\mathcal{L}_c(t) = \tfrac{1}{2}\left((1 - \mu\alpha_c(t))^2 + \sigma_x^2 \sum_{j=1}^K \alpha_j^2(t)\right). \tag{3}$$

For any given $t$, the class-balanced loss gap is $\Delta\mathcal{L}_{\text{bal}}(t) := \sum_c \mathcal{L}_c^{\text{GD}}(t) - \mathcal{L}_c^{\text{Spec}}(t)$.

In order to bound this gap, we start by computing its derivative

$$\Delta\dot{\mathcal{L}}_{\text{bal}}(t) = -\tfrac{1}{k}\sum_c(1 - \mu\alpha_c^{\text{GF}}(t))\mu\dot{\alpha}_c^{\text{GF}}(t) + \sigma_x^2\sum_c \alpha_c^{\text{GF}}(t)\dot{\alpha}_c^{\text{GF}}(t)$$
$$+ \tfrac{1}{k}\sum_c(1 - \mu\alpha_c^{\text{Spec}}(t))\mu\dot{\alpha}_c^{\text{Spec}}(t) - \sigma_x^2\sum_c \alpha_c^{\text{Spec}}(t)\dot{\alpha}_c^{\text{Spec}}(t).$$

Next, from the expression of $\alpha_c^{\text{GF}}(t)$ in Eq. (2) and $\dot{\alpha}_m^{\text{GF}}(t) = \mu p_m e^{-s_m^{xx}t}$, we use $\alpha_m^{\text{GF}}(t)\dot{\alpha}_m^{\text{GF}}(t) \geq 0$, and $\dot{\alpha}_m^{\text{GF}}(t) \leq \mu p_m$. Combining with $\alpha_m^{\text{Spec}}(t)$ from Proposition 1 and $\dot{\alpha}_m^{\text{Spec}}(t) = 1$ for $t \leq t^*$, we have:

$$\Delta\dot{\mathcal{L}}_{\text{bal}}(t) = -\tfrac{1}{k}\mu^2 + (1 - \mu t)\mu - \sigma_x^2 kt.$$

Next, we use $t \leq t^* = \alpha_m$, and using the assumption on $p_m$ in the definition of $\alpha_m$, we get $\Delta\dot{\mathcal{L}}_{\text{bal}}(t) \geq \mu/2$. This is then integrated over $t$ to obtain the final result.

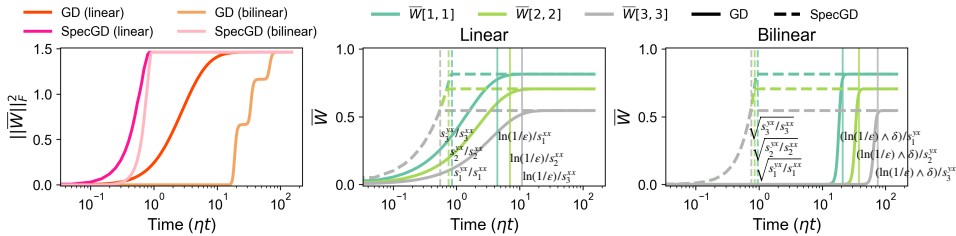

Figure 5: Comparison of the dynamics of the squared norm of singular value matrix of the iterates, and the individual singular values, for GD and SpecGD across linear ($L=1$) and bilinear ($L=2$) models for class-imbalanced data (priors $p_1 > p_2 > p_3$). In both cases, GD learns more dominant spectral components first, whereas SpecGD learns all components at the same rate. The gap between saturation of different components for SpecGD decreases as model depth increases from 1 to 2, as proved in Proposition 2.

The proof of Theorem 2 is based on a similar idea. The key distinction is that we don't have the precise dynamics of NGD(F) iterates. Specifically, the NGF update for $c \in [k]$ is written as

$$\dot{\alpha}_c^{\text{NGF}}(t) = \frac{r_c(t)}{R(t)}, \quad \text{where } r_c(t) := s_c^{\text{yx}} - s_c^{\text{xx}} \alpha_c^{\text{NGF}}(t) \text{ and } R(t) := \sqrt{\sum_{c=1}^{k} r_c^2(t)}.$$

Then, we have that $0 \leq r_c(t) \leq s_c^{\text{yx}}$, and using this with $R(t) \geq \max_c r_c(t)$ gives $\dot{\alpha}_c^{\text{NGF}}(t) \leq 1$, and hence $\alpha_c^{\text{NGF}}(t) \leq t$. Using these we lower bound $\Delta \mathcal{L}'_m(t)$ and $\Delta \mathcal{L}'_{\text{bal}}(t)$, and integrate over $t$.

### 3.3 EFFECT OF DEPTH

To investigate how increasing model depth $L$ affects SpecGD dynamics, consider deep linear model $\boldsymbol{W} := \prod_{i=0}^{L-1} \boldsymbol{W}^{L-1-i}$ with total loss $\mathcal{L}(\boldsymbol{W}^0, \ldots, \boldsymbol{W}^{L-1}) := \frac{1}{2}\mathbb{E}\|\boldsymbol{y} - \prod_{i=0}^{L-1} \boldsymbol{W}^{L-1-i}\boldsymbol{x}\|_2^2$.

**Bilinear Model.** We begin with the case $L = 2$, which is particularly instructive due to its connection to the Unconstrained Features Model (UFM). The UFM is widely used to study the geometry of learned representations in deep and nonlinear networks under finite-sample regimes (Yang et al., 2018; Mixon et al., 2020). It serves as a proxy for architectures with sufficient capacity to freely optimize both the feature representation ($\boldsymbol{W}^0\boldsymbol{x}$) and the classification head ($\boldsymbol{W}^1$). The bilinear model recovers this framework when the input features $\boldsymbol{x}$ are orthogonal, making it a natural setting for analyzing the interplay between feature learning and classifier dynamics.

The following result characterizes the SpecGD dynamics of $\boldsymbol{W}_t = \boldsymbol{W}_t^1 \boldsymbol{W}_t^0$.

**Proposition 2** (Simplified)**.** *Suppose Condition 1 holds. Let the weights be initialized as $\boldsymbol{W}_0^0 = e^{-\delta}\boldsymbol{Q}[\boldsymbol{I}_{d_1} \ \boldsymbol{0}_{d-d_1}]\boldsymbol{V}^\top$ and $\boldsymbol{W}_0^1 = e^{-\delta}\boldsymbol{U}[\boldsymbol{I}_k \ \boldsymbol{0}_{d_1-k}]\boldsymbol{Q}^\top$, where $\boldsymbol{Q} \in \mathbb{R}^{d_1 \times d_1}$ is an orthonormal matrix, $k < d_1 < d$, and $\delta > 0$ is a constant. Then, for SpecGD with, at each iteration, $\boldsymbol{W}_t^0 := \boldsymbol{Q}\overline{\boldsymbol{W}}_t^0 \boldsymbol{V}^\top$, $\boldsymbol{W}_t^1 = \boldsymbol{U}\overline{\boldsymbol{W}}_t^1 \boldsymbol{Q}^\top$, where $\overline{\boldsymbol{W}}_t^0, \overline{\boldsymbol{W}}_t^1$ are zero except their main diagonal along which, in the limit $\delta \to \infty$, they evolve as follows for $c \in [k]$:*

$$\overline{\boldsymbol{W}}_t^0[c,c] = \overline{\boldsymbol{W}}_t^1[c,c] = \eta t \mathbb{1}\left[t \leq \frac{1}{\eta}\sqrt{\frac{s_c^{\text{yx}}}{s_c^{\text{xx}}}}\right] + \sqrt{\frac{s_c^{\text{yx}}}{s_c^{\text{xx}}}}\mathbb{1}\left[t > \frac{1}{\eta}\sqrt{\frac{s_c^{\text{yx}}}{s_c^{\text{xx}}}}\right].$$

For clarity, we consider $\delta \to \infty$ in the above proposition to state the SpecGD dynamics. See App. B.5 for the complete version (for any $\delta > 0$) and the proof.

We find that, in contrast to the linear model where SpecGD learns the $c^{\text{th}}$ component in time $t_c = \frac{1}{\eta}\frac{s_c^{\text{yx}}}{s_c^{\text{xx}}}$, for the bilinear model, it saturates in time $t_c \approx \frac{1}{\eta}\sqrt{\frac{s_c^{\text{yx}}}{s_c^{\text{xx}}}}$ (for large $\delta$). This is quite different from the effect of increasing model depth on the dynamics of GD. As discussed in Saxe et al. (2013); Gidel et al. (2019), for GD, the time to reach $\epsilon$-close to the optimal (for the $c^{\text{th}}$ component) is $t_c = \frac{\ln(\epsilon)}{\ln(1-\eta s_c^{\text{xx}})} \approx \frac{\ln(1/\epsilon)}{\eta s_c^{\text{xx}}}$ (for small $\eta$) for a linear model, compared to $t_c \approx \frac{\ln(1/\epsilon-1)+\ln(s_c^{\text{xx}}e^{\delta}/s_c^{\text{yx}}-1)}{2\eta s_c^{\text{yx}}} \approx \frac{\ln(1/\epsilon)+\delta}{2\eta s_c^{\text{yx}}}$ (for small $\epsilon$ and large $\delta$) for a bilinear model. To make the comparison more concrete, recall from Lemma 3.1 that for our class-imbalanced setting with data

model (DM), when $\mu = 1$, $s_c^{xx} = p_c + \text{SNR}^{-1}$ and $s_c^{yx} = p_c$. The GD dynamics for linear vs. bilinear model depend on the SNR: when SNR is high, we observe the stage-wise learning dynamics in both cases, *i.e.*, spectral components are learned in descending order of class priors. However, when SNR is low, these stage-wise dynamics may not be apparent for the linear model as $t_c \propto \frac{1}{p_c + \text{SNR}^{-1}} \approx \text{SNR}$.

In contrast, for SpecGD, $t_c \propto \left(\frac{p_c}{p_c + \text{SNR}^{-1}}\right)^{1/L}$, where $L \in \{1, 2\}$. This means the dynamics are qualitatively similar for both models; however, the gap between saturation times on different components is smaller for the deeper bilinear model.

We illustrate these differences between GD and SpecGD and the effect of increasing model depth in Fig. 5, where similar to Fig. 2, we consider a class-imbalanced setting with $\eta = 0.05$, $d = k = 3$, $\mu = 1$, $\sigma_x^2 = 0.125$, and $p_1, p_2, p_3$ set as $0.55, 0.3, 0.15$, respectively. We set $\epsilon = 0.05, \delta = 10$.

Next, we investigate whether for SpecGD, the gap between saturation on different components becomes smaller as model depth is increased further.

**Deep Linear Model.** We next consider deeper models with $L \geq 2$. The dynamics of $W_t = \prod_{i=0}^{L-1} W_t^{L-1-i}$ for SpecGD can be characterized in a similar way as Proposition 2. Specifically, $\{\overline{W}_t^l\}_{l=0}^{L-1}$ defined analogously remain diagonal with entries $c \in [k]$, in the limit $\delta \to \infty$, evolving as:

$$\overline{W}_t^l[c, c] = \eta t \mathbb{1}\left[t \leq \frac{1}{\eta}\left(\frac{s_c^{yx}}{s_c^{xx}}\right)^{\frac{1}{L}}\right] + \left(\frac{s_c^{yx}}{s_c^{xx}}\right)^{\frac{1}{L}} \mathbb{1}\left[t > \frac{1}{\eta}\left(\frac{s_c^{yx}}{s_c^{xx}}\right)^{\frac{1}{L}}\right].$$

See App. B.6 for a formal statement and the proof. We can quantify the gap between saturation of the minority vs. majority component as $\Delta T := \frac{t_{\max} - t_{\min}}{t_{\min}} = \left(\frac{s_1^{yx}/s_1^{xx}}{s_k^{yx}/s_k^{xx}}\right)^{1/L} - 1$. For the class-imbalanced setting in Eq. (DM), using Lemma 3.1, $\Delta T = \left(\frac{\text{SNR} + 1/p_m}{\text{SNR} + 1/p_M}\right)^{1/L} - 1$.

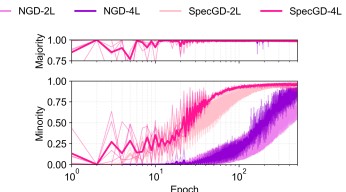

Figure 6: Test accuracy dynamics of NGD and SpecGD for a 2-layer vs. 4-layer MLP on the Colored-MNIST dataset.

We find that as $L$ increases, the gap between the saturation of the different components becomes smaller. We empirically validate this result for SpecGD in Fig. 6 by comparing the test accuracy dynamics of 2-layer and 4-layer MLPs on the Colored-MNIST dataset. A similar trend is observed for NGD, where depth speeds up learning of the minority component. This is an interesting empirical parallel, given that precise theoretical dynamics for GD have primarily been characterized for depth $L = 2$ (Gidel et al., 2019; Saxe et al., 2013).

## 4 EXPERIMENTAL RESULTS

In this section, we present experimental results on datasets with group and class imbalance, comparing the performance of Muon to SGD. Our analysis is guided by the spectral perspective developed in Sec. 3.2. For class imbalance, recall that spectral components correspond to class priors. We extend this analogy to group imbalance, where inputs contain both core and spurious features, and groups are defined by different core-spurious feature combinations. As seen in our motivating Colored-MNIST example (Fig. 1), spurious features (e.g., color) act as the dominant spectral component, while core features (e.g., digit shape) are less dominant (see also (Ng et al., 2024), which demonstrates this for some other datasets with spurious correlations). Our experiments show that Muon promotes a more balanced learning of different spectral components, thereby improving generalization relative to SGD. Finally, while our theory focused on the dynamics of SpecGD versus (N)GD, our empirical study also includes Adam, offering a broader comparison with a widely used adaptive optimizer.

**Group Imbalance Datasets.** Here, we consider three widely used datasets with group imbalance.

*Dominoes Dataset.* We consider the MNIST-CIFAR dataset from the Dominoes benchmark (Shah et al., 2020; Pagliardini et al., 2022), where each image is generated by stacking an MNIST digit from class $\{0, 1\}$ on the top with a CIFAR-10 image from class {automobile, truck} on the bottom. The MNIST part is spuriously correlated with the label $95\%$ of the time, while the CIFAR part is $100\%$ predictive of the label. Example images are shown in Fig. 7 (left).

We train a ResNet-34 model using SGD, Adam, Shampoo, and Muon and compare their generalization using two metrics on the test set: the worst-group accuracy and the decoded worst-group accuracy (see App. D.3 for details). The latter is obtained by freezing the learned representation of the trained model, re-training only the final linear layer with logistic regression on a group-balanced validation set, and then evaluating its performance on the test set. This measures

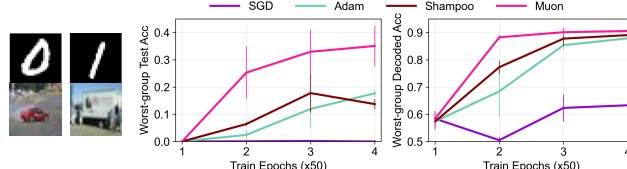

Figure 7: Example images from the MNIST-CIFAR dataset (left), and comparison of worst-group test accuracy (middle) and worst-group decoded test accuracy (right) dynamics of SGD, Adam, Shampoo and Muon (see text for discussion). *Muon learns core features faster and has superior worst-group performance while SGD relies on spurious features.*

the extent to which the model has learned the core feature information in its representations. From Fig. 7, we see that Muon consistently outperforms SGD across both metrics throughout training. Compared to Adam, Muon shows greater gains early in training, eventually attaining a similar performance. Muon, as well as Shampoo and Adam, achieve much higher decoded accuracy than SGD, which suggests that they successfully learn the core features (which correspond to less dominant spectral components), whereas SGD struggles to do so.

***Subgroup Robustness Benchmarks.*** Next, we consider two benchmark subgroup robustness datasets: MultiNLI (Williams et al., 2018) and CelebA (Liu et al., 2015). MultiNLI consists of sentence pairs, where the task is to classify the relationship of the second sentence to the first as *entailment*, *neutral*, or *contradiction*. The spurious feature is the presence of *negation words*, which often indicate contradiction. CelebA contains images of celebrity faces, with the task of predicting whether the hair color is *blonde/not blonde*, and the celebrity's gender (*male/female*) serves as the spurious attribute.

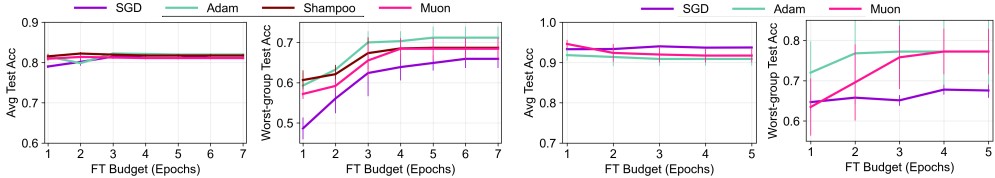

Figure 8: Comparison of the worst-group and average test accuracy over the number of fine-tuning (FT) epoch budget (see text for details) for different optimizers on the (left) MultiNLI and (right) CelebA datasets. The average test accuracy is comparable across all optimizers. For the worst-group accuracy, spectrum-aware optimizers (Muon and Shampoo) consistently outperform SGD. However, as compared to Adam, for MultiNLI, Muon is slightly worse, while for CelebA, its performance becomes comparable for more FT epochs.

We fine-tune a pretrained BERT `bert-base-uncased` model (Devlin et al., 2019) on the MultiNLI dataset, and an ImageNet-pretrained ResNet-50 model on CelebA, using different optimizers and evaluate the average (group-balanced) test accuracy and the worst-group accuracy over the number of finetuning epochs. To ensure our comparisons are robust, we conduct extensive hyperparameter sweeps for all optimizers (see App. D.4 for details). We select the best hyperparameter configuration for any given optimizer and number of fine-tuning epochs, $T$, based on the highest average worst-group accuracy achieved on the validation set, averaged across multiple initialization seeds, at any point up to $T$ epochs. We then report the performance for the optimal configuration for different FT epochs. From Fig. 8, we see that Muon outperforms SGD on both datasets. On MultiNLI, Adam attains better performance than Muon/Shampoo whereas on CelebA, its performance is comparable to Muon. These experiments show that Muon promotes more balanced learning of spectral components than SGD, thereby improving generalization.

**Class-imbalanced Datasets.** Here, we extend our investigation of class imbalance to language modeling. While our previous experiments and theory focused on one-hot classification, we now shift our comparison of optimizers to a next-token prediction (NTP) task. This task presents a natural imbalance over the distribution of tokens, as word frequencies follow a long-tail distribution described by Zipf's law (Piantadosi, 2014).

***TinyStories Dataset.*** To examine how different optimizers compare on rare token learning, we train small Transformer models on the TinyStories (Eldan & Li, 2023) corpus and compare the performance of Muon, SGD, and Adam. We group tokens into "frequent" and "rare" categories based on their corpus statistics and evaluate the token-level accuracy for each group separately on a held-out test set. Complete training details are available in App. D.5. The performance on rare and frequent tokens is reported in Fig. 9. Consistent with our findings in the one-hot class-imbalanced classification, we observe that Muon generalizes faster on both the majority (frequent)

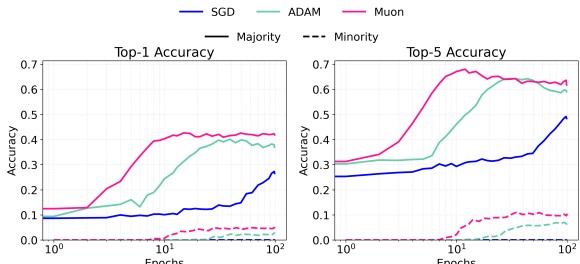

Figure 9: Performance comparison of SGD, Adam, and Muon on frequent (majority) vs. rare (minority) tokens, evaluated by Top-1 and Top-5 accuracy on a validation set of TinyStories. The results demonstrate that Muon achieves faster generalization on both frequent and rare tokens compared to SGD and Adam.

and minority (rare) components compared to SGD, and in this case, even compared to Adam.

***Attribute-Organism Classification.*** To extend our findings from statistical imbalance to a setting with explicit conceptual structure, we conduct a toy experiment on hierarchical concept learning (Zhao & Thrampoulidis, 2025). In this task, a model must classify organisms (e.g., "dog") based on their attributes, which are shared at different levels of a taxonomic tree (e.g., "needs oxygen" for all animals). We observe distinct learning dynamics: GD learns the coarse, high-level categories (plant vs. animal) first, while Muon exhibits a more balanced learning progression across both coarse and fine-grained concepts (see App. D.6 for details)

## 5 DISCUSSION AND OUTLOOK

Our work investigates how the spectral design of optimizers like Muon and Shampoo yields generalization benefits, using imbalanced data as a testbed. The core mechanism we identify is that their canonical form, SpecGD, implicitly learns all principal components of the data at an equal rate, in stark contrast to GD.

A crucial test of this insight is how its practical variants, Muon and Shampoo, perform against strong, widely-used baselines like Adam. In our experiments in Sec. 4, we find that Adam's generalization performance is only slightly worse—or sometimes even better—than that of Muon and Shampoo. This is consistent with recent large-scale benchmarks for LLM pretraining, which show that well-tuned Adam variants remain highly competitive (Wen et al., 2025; Semenov et al., 2025). Previous work has used different perspectives to explain Adam's superiority over (S)GD: i) Kunstner et al. (2024) show how heavy-tailed class-imbalanced settings lead to gains for Adam by focusing on their *optimization* dynamics; ii) Vasudeva et al. (2025b) show that Adam promotes richer feature learning than SGD in neural networks, which can lead to better generalization in group-imbalanced settings. Together, these works underscore Adam's advantages in imbalanced regimes, albeit through different mechanisms. Our motivation for using this testbed stems from recognizing that spectral components of imbalanced data directly correspond to majority and minority classes or groups, and our analysis reveals that SpecGD achieves its gains through a fundamentally different and interpretable mechanism—balanced learning of principal components.

A natural question is how the implicit balancing effect of spectral methods compares to explicitly designed techniques like loss re-weighting (Byrd & Lipton, 2019; Cao et al., 2019; Sagawa et al., 2019; Menon et al., 2020; Liu et al., 2021). In App. D.8, we show that while weighted cross-entropy (wCE) effectively encourages balanced learning of spectral components on NMD, Muon provides a similar, though weaker, effect "for free" without requiring explicit group priors. This suggests spectral methods could be useful when such information is unavailable for re-weighting.

Our work provides a framework to understand the generalization of spectrum-aware optimizers within a controlled setting using population statistics and squared loss. A natural next step is to build upon this foundation by extending the analysis to more complex regimes using finite samples and cross-entropy (CE) loss. To bridge this gap, we conduct an initial empirical analysis of rate of learning spectral components for nonlinear models trained with CE loss, by tracking singular values of the logit matrices during training (see App. D.10 for details). We outline future directions and their associated challenges in App. F.

## ACKNOWLEDGMENTS

The authors acknowledge use of the Discovery cluster by USC CARC and the Sockeye cluster by UBC Advanced Research Computing. This work was partially funded by the NSERC Discovery Grant No. 2021-03677 and the Alliance Grant ALLRP 581098-22. PD and YZ were also supported by the UBC 4YF Doctoral Fellowship. This work was also supported in part by NSF CAREER Award CCF-2239265, an Amazon Research Award, a Google Research Scholar Award and an Okawa Foundation Research Grant.

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

## APPENDIX

## A  OPTIMIZERS

In this section, we list the update rules for all the optimizers considered in the paper for completeness. We start with the update rules of NGD, SignGD, and SpecGD, then list their momentum versions (NMD, Signum, and Muon), and then write the updates for Shampoo and Adam.

Using the notation introduced in Sec. 3, updates are of the form

$$\boldsymbol{W}_{t+1} = \boldsymbol{W}_t - \eta \boldsymbol{\Delta}_t, \ \text{ where } \boldsymbol{\Delta}_t := \mathrm{argmax}_{\|\boldsymbol{\Delta}\| \leq 1} \langle \boldsymbol{\nabla}_t, \boldsymbol{\Delta} \rangle. \tag{4}$$

**NGD update.**  For NGD, the arg max in Eq. (4) uses $\|\cdot\|_F$, and we get $\boldsymbol{\Delta}_t = \dfrac{\boldsymbol{\nabla}_t}{\|\boldsymbol{\nabla}_t\|_F}$.

**SignGD update.** For signGD, the arg max in Eq. (4) uses max, and we get $\boldsymbol{\Delta}_t = \text{sign}(\boldsymbol{\nabla}_t)$, where $\text{sign}(x) := \frac{x}{|x|}$ and $\text{sign}(0) = 0$, applied element-wise on the matrix $\boldsymbol{\nabla}_t$.

**SpecGD update.** For SpecGD, the arg max in Eq. (4) uses the spectral norm. Let the truncated SVD of $\boldsymbol{\nabla}_t$ be $\boldsymbol{U}_t \boldsymbol{\Sigma}_t \boldsymbol{V}_t^\top$, where $\boldsymbol{U}_t$ and $\boldsymbol{V}_t$ are orthonormal matrices and $\boldsymbol{\Sigma}_t$ is a diagonal matrix with positive diagonal entries. Using this, we have $\boldsymbol{\Delta}_t = \boldsymbol{U}_t \boldsymbol{V}_t^\top$.

For NMD, Signum, and Muon (with exact matrix operations), the updates mirror NGD, SignGD, and SpecGD, respectively, except that they use the momentum term $\boldsymbol{M}_t = \beta \boldsymbol{M}_{t-1} + (1 - \beta)\overline{\boldsymbol{\nabla}}_t$ in place of the raw gradient.

**Shampoo update.** For Shampoo, first define the preconditioning matrices

$$\boldsymbol{L}_t = \beta_2 \boldsymbol{L}_{t-1} + (1 - \beta_2)\boldsymbol{\nabla}_t \boldsymbol{\nabla}_t^\top \quad \text{and} \quad \boldsymbol{R}_t = \beta_2 \boldsymbol{R}_{t-1} + (1 - \beta_2)\boldsymbol{\nabla}_t^\top \boldsymbol{\nabla}_t,$$

where the parameter $\beta_2$ denotes the preconditioning accumulation parameter, and the momentum matrix $\boldsymbol{M}_t = \beta_1 \boldsymbol{M}_{t-1} + (1 - \beta_1)\boldsymbol{\nabla}_t$. Using these preconditioners, the update is $\boldsymbol{\Delta}_t = \boldsymbol{L}_t^{-1/4} \boldsymbol{M}_t \boldsymbol{R}_t^{-1/4}$.

It is easy to see that Shampoo with exact matrix operations reduces to SpecGD when we set $\beta_1 = \beta_2 = 0$ as follows. Let the SVD of $\boldsymbol{\nabla}_t$ be $\boldsymbol{U}_t \boldsymbol{\Sigma}_t \boldsymbol{V}_t^\top$. Then, since $\beta_2 = 0$, the preconditioners are $\boldsymbol{L}_t = \boldsymbol{U}_t \boldsymbol{\Sigma}_t^2 \boldsymbol{U}_t^\top$ and $\boldsymbol{R}_t = \boldsymbol{V}_t \boldsymbol{\Sigma}_t^2 \boldsymbol{V}_t^\top$. Using these, since $\beta_1 = 0$, the update is $\boldsymbol{\Delta}_t = \boldsymbol{U}_t \boldsymbol{\Sigma}_t^{-1/2}(\boldsymbol{U}_t^\top \boldsymbol{U}_t)\boldsymbol{\Sigma}_t(\boldsymbol{V}_t^\top \boldsymbol{V}_t)\boldsymbol{\Sigma}_t^{-1/2}\boldsymbol{V}_t^\top = \boldsymbol{U}\boldsymbol{V}^\top$, which is same as the SpecGD update.

**Adam update.** For Adam, let $\hat{\boldsymbol{M}}_t = \frac{\boldsymbol{M}_{t+1}}{1-\beta_1^{t+1}} = \frac{1}{1-\beta_1^{t+1}}\left(\beta_1 \boldsymbol{M}_t + (1-\beta_1)\boldsymbol{\nabla}_t\right)$ denote the bias-corrected first-moment estimate, and $\hat{\boldsymbol{Z}}_t = \frac{\boldsymbol{Z}_{t+1}}{1-\beta_2^{t+1}} = \frac{1}{1-\beta_2^{t+1}}\left(\beta_2 \boldsymbol{Z}_t + (1-\beta_2)\boldsymbol{\nabla}_t \odot \boldsymbol{\nabla}_t\right)$ denote the bias-corrected second (raw) moment estimate, where $\odot$ denotes the Hadamard product, and $\beta_1, \beta_2$ denote the momentum parameters.

Then, the update is $\boldsymbol{\Delta}_t = (\hat{\boldsymbol{Z}}_t + \epsilon \mathbf{1}\mathbf{1}^\top)^{\circ-1/2} \odot \hat{\boldsymbol{M}}_t$, where $(\cdot)^\circ$ denotes the Hadamard power, $\epsilon > 0$ is the numerical precision parameter, and $\mathbf{1}$ denotes the all-ones vector. It is easy to see that Adam reduces to SignGD when we set $\beta_1 = \beta_2 = \epsilon = 0$.

# B PROOFS

## B.1 PROOF OF LEMMA 3.1

The covariance matrix is

$$\begin{aligned}
\boldsymbol{\Sigma}_{\boldsymbol{x}} &:= \mathbb{E}\big[\boldsymbol{x}\boldsymbol{x}^\top\big] = \mathbb{E}_{y,\boldsymbol{\varepsilon}}\Big[(\boldsymbol{\mu}_y + \boldsymbol{\varepsilon})(\boldsymbol{\mu}_y + \boldsymbol{\varepsilon})^\top\Big] \\
&= \underbrace{\mathbb{E}_y\big[\boldsymbol{\mu}_y \boldsymbol{\mu}_y^\top\big]}_{=:\boldsymbol{\Sigma}_{\boldsymbol{\mu}}} + \mathbb{E}_y\big[\boldsymbol{\mu}_y\big]\mathbb{E}_{\boldsymbol{\varepsilon}}\big[\boldsymbol{\varepsilon}^\top\big] + \mathbb{E}_{\boldsymbol{\varepsilon}}\big[\boldsymbol{\varepsilon}\big]\mathbb{E}_y\big[\boldsymbol{\mu}_y^\top\big] + \mathbb{E}\big[\boldsymbol{\varepsilon}\boldsymbol{\varepsilon}^\top\big] = \boldsymbol{\Sigma}_{\boldsymbol{\mu}} + \sigma_x^2 \boldsymbol{I}_d,
\end{aligned}$$

since $\boldsymbol{\varepsilon}$ and $y$ are independent and $\mathbb{E}[\boldsymbol{\varepsilon}] = 0$. Further,

$$\boldsymbol{\Sigma}_{\boldsymbol{\mu}} = \sum_{c=1}^k p_c \boldsymbol{\mu}_c \boldsymbol{\mu}_c^\top = \boldsymbol{M}\boldsymbol{P}\boldsymbol{M}^\top, \qquad \boldsymbol{M} := \begin{bmatrix} \bar{\boldsymbol{\mu}}_1 \, \bar{\boldsymbol{\mu}}_2 \, \cdots \, \bar{\boldsymbol{\mu}}_k \end{bmatrix} \in \mathbb{R}^{d\times K}, \; \boldsymbol{P} := \mu^2 \, \text{diag}(p_1, \ldots, p_k).$$

Then, we can write $\boldsymbol{\Sigma}_{\boldsymbol{x}} = \boldsymbol{V}\boldsymbol{S}_{\boldsymbol{xx}}\boldsymbol{V}^\top$, where

$$\boldsymbol{S}_{\boldsymbol{xx}} = \text{diag}(\mu^2 p_1 + \sigma_x^2, \ldots, \mu^2 p_k + \sigma_x^2, \underbrace{\sigma_x^2, \ldots, \sigma_x^2}_{d-k \text{ times}}), \tag{5}$$

$$\boldsymbol{V} = \begin{bmatrix} \boldsymbol{M} \; \boldsymbol{V}_\perp \end{bmatrix} = \begin{bmatrix} \bar{\boldsymbol{\mu}}_1, \ldots, \bar{\boldsymbol{\mu}}_k, \boldsymbol{v}_{k+1}, \ldots, \boldsymbol{v}_d \end{bmatrix} \in \mathbb{R}^{d\times d}, \quad \{\boldsymbol{v}_i\}_{i=k+1}^d \perp \{\boldsymbol{\mu}_c\}_{c=1}^k.$$

Here, we used the assumption on orthonormality of the means.

The cross-covariance is

$$\boldsymbol{\Sigma}_{\boldsymbol{yx}} := \mathbb{E}\big[\boldsymbol{y}\,\boldsymbol{x}^\top\big] = \sum_{c=1}^k p_c \, \boldsymbol{y}_c \, \mathbb{E}\big[\boldsymbol{x}^\top \mid y = c\big] = \sum_{c=1}^k \mu p_c \, \boldsymbol{e}_c \, \bar{\boldsymbol{\mu}}_c^\top.$$

We can write $\mathbf{\Sigma_{yx}} = \mathbf{U}\,\mathbf{S_{yx}}\,\mathbf{V}^\top$, where

$$U = I_k, \quad S_{yx} = [\mu \operatorname{diag}(p_1, \ldots, p_k) \quad \mathbf{0}_{d-k}]. \tag{6}$$

Note that $\mathbf{S_{yx}}$ is a $k \times d$ matrix, where class priors $p_i$, $i \in [k]$ denote entries along the main diagonal with zeros in the rest of the entries. For brevity, we will denote this as $\mathbf{S_{yx}} = \mu \operatorname{diag}(p_1, \ldots, p_k)_{k \times d}$ from here on.

## B.2 PROOF OF PROPOSITION 1

For completeness, we start by deriving the discrete-time evolution for GD.

We can write the gradient as

$$\nabla \mathcal{L}(\mathbf{W}_t) = -\mathbb{E}\left[(\mathbf{y} - \mathbf{W}_t \mathbf{x})\mathbf{x}^\top\right] = -\mathbf{U}\mathbf{S_{yx}}\mathbf{V}^\top + \mathbf{W}_t \mathbf{V}\mathbf{S_{xx}}\mathbf{V}^\top.$$

For GD, we have

$$
\begin{aligned}
\mathbf{W}_{t+1} &= \mathbf{W}_t - \eta \nabla \mathcal{L}(\mathbf{W}_t) \\
&= \mathbf{W}_t + \eta \mathbf{U}\mathbf{S_{yx}}\mathbf{V}^\top - \eta \mathbf{W}_t \mathbf{V}\mathbf{S_{xx}}\mathbf{V}^\top = \mathbf{W}_t\big(\mathbf{I} - \eta \mathbf{V}\mathbf{S_{xx}}\mathbf{V}^\top\big) + \eta \mathbf{U}\mathbf{S_{yx}}\mathbf{V}^\top \\
&= \mathbf{W}_0\big(\mathbf{I} - \eta \mathbf{V}\mathbf{S_{xx}}\mathbf{V}^\top\big)^{t+1} + \sum_{\tau=0}^{t} \eta \mathbf{U}\mathbf{S_{yx}}\big(\mathbf{I} - \eta \mathbf{S_{xx}}\big)^\tau \mathbf{V}^\top.
\end{aligned}
$$

Since $\mathbf{W}_0 = \mathbf{0}$, this gives

$$\overline{\mathbf{W}}_{t+1} := \mathbf{U}^\top \mathbf{W}_{t+1}\mathbf{V} = \eta \mathbf{S_{yx}} \sum_{\tau=0}^{t}\big(\mathbf{I} - \eta \mathbf{S_{xx}}\big)^\tau.$$

Assuming $\eta < \frac{1}{d_{\max}}$, we get

$$\overline{W}_{t+1}[i,i] = \eta s_i^{yx} \sum_{\tau=0}^{t}\big(1 - \eta\,s_i^{xx}\big)^\tau = s_i^{yx}\,\frac{1 - \big(1 - \eta s_i^{xx}\big)^{t+1}}{s_i^{xx}}.$$

For sufficiently small $\eta$, this gives the approximation

$$\overline{W}_{t+1}[i,i] \approx \frac{s_i^{yx}}{s_i^{xx}}\big(1 - e^{-\eta s_i^{xx}(t+1)}\big).$$

For SpecGD, note that the gradient can be written in terms of $\overline{\mathbf{W}}_t$ as

$$\mathbf{\nabla}_t = \nabla \mathcal{L}(\mathbf{W}_t) = -\mathbf{U}\mathbf{S_{yx}}\mathbf{V}^\top + \mathbf{W}_t \mathbf{V}\mathbf{S_{xx}}\mathbf{V}^\top = \mathbf{U}(\mathbf{S_{xx}}\overline{\mathbf{W}}_t - \mathbf{S_{yx}})\mathbf{V}^\top.$$

Starting at $\mathbf{W}_t = 0 \Leftrightarrow \overline{\mathbf{W}}_t = 0$ gives $\mathbf{\nabla}_0 = -\mathbf{U}\mathbf{S_{yx}}\mathbf{V}^\top$.

Thus, $\mathbf{W}_1 = \eta \mathbf{U}\mathbf{V}_k^\top \Rightarrow \overline{W}_1[i,j] = \eta \begin{cases} 1 & i = j \in [k] \\ 0 & i \neq j \end{cases}$. Here, $\mathbf{V}_k \in \mathbb{R}^{d \times k}$ denotes the first $k$ columns of $\mathbf{V}$. Proceeding this way, we arrive at the following update rule for all $t$,

$$\mathbf{W}_{t+1} = \mathbf{W}_t + \eta \sum_{i:\overline{W}_t[i,i]\, s_i^{xx} < s_i^{yx}} \mathbf{u}_i \mathbf{v}_i^\top.$$

This gives

$$\overline{\mathbf{W}}_{t+1} = \overline{\mathbf{W}}_t + \eta \sum_{i:\ \overline{W}_t[i,i]\, s_i^{xx} < s_i^{yx}} \mathbf{e}_i \mathbf{e}_i^\top.$$

Since $\mathbf{W}_0 = \mathbf{0}$, we conclude with the desired:

$$\overline{W}_{t+1}[i,i] = \eta\,(t+1)\,\mathbb{1}\left[(t+1) \leq \frac{s_i^{yx}}{\eta s_i^{xx}}\right] + \frac{s_i^{yx}}{s_i^{xx}}\,\mathbb{1}\left[(t+1) > \frac{s_i^{yx}}{\eta s_i^{xx}}\right].$$

For the continuous time versions, aka gradient flow (GF) and spectral gradient flow (specGF), the steps are very similar:

**Gradient Flow (GF).** We first state the GF equation

$$\frac{d}{dt} \boldsymbol{W}(t) = -\nabla \mathcal{L}(\boldsymbol{W}(t)).$$

Using this, we can write the evolution of $\overline{\boldsymbol{W}}(t) := \boldsymbol{U}^\top \boldsymbol{W}(t) \boldsymbol{V}$ as

$$\frac{d}{dt} \overline{\boldsymbol{W}}(t) = -\overline{\boldsymbol{W}}(t) \boldsymbol{S_{xx}} + \boldsymbol{S_{yx}}, \qquad \overline{\boldsymbol{W}}(0) = 0,$$

$$\implies \quad \overline{\boldsymbol{W}}(t) = \boldsymbol{S_{xx}}^{-1} \big( \boldsymbol{I} - e^{-t \boldsymbol{S_{xx}}} \big) \boldsymbol{S_{yx}},$$

or entrywise

$$\overline{\boldsymbol{W}}(t)[i,i] \;=\; \frac{s_i^{\text{yx}}}{s_i^{\text{xx}}} \big( 1 - e^{-t s_i^{\text{xx}}} \big).$$

**Spectral Gradient Flow (SpecGF).** We first state SpecGF equation

$$\frac{d}{dt} \boldsymbol{W}(t) = -\boldsymbol{U}(t) \boldsymbol{V}(t)^\top,$$

where the gradient has SVD $\nabla \mathcal{L}(\boldsymbol{W}(t)) = \boldsymbol{U}(t) \boldsymbol{\Sigma}(t) \boldsymbol{V}(t)^\top$. Following the same steps as for SpecGD, we can show that $\boldsymbol{U}(t) = \boldsymbol{U}$, $\boldsymbol{V}(t) = \boldsymbol{V}$, and $\overline{\boldsymbol{W}}(t)$ evolves as

$$\frac{d}{dt} \overline{\boldsymbol{W}}(t)[i,i] \;=\; \begin{cases} 1 & \text{if } \overline{\boldsymbol{W}}(t)[i,i] s_i^{\text{xx}} < s_i^{\text{yx}}, \\ 0 & \text{if } \overline{\boldsymbol{W}}(t)[i,i] s_i^{\text{xx}} = s_i^{\text{yx}}, \end{cases}$$

with solution

$$\overline{\boldsymbol{W}}(t)[i,i] = t \, \mathbb{1}\left[ t \le \frac{s_i^{\text{yx}}}{s_i^{\text{xx}}} \right] + \frac{s_i^{\text{yx}}}{s_i^{\text{xx}}} \, \mathbb{1}\left[ t > \frac{s_i^{\text{yx}}}{s_i^{\text{xx}}} \right].$$

### B.3  PROOF OF THEOREM 1

We first state the full version of Theorem 1 below followed by its proof.

**Theorem 3.** *Assume data model* (DM) *and zero initialization. Let $t^\star = \frac{s_m^{\text{yx}}}{s_m^{\text{xx}}}$ be the first time SpecGF fits the minority class $m$. Further assume $\mu \ge 1$ and $p_m \le \frac{1}{3SNR + 4k}$. Then for every $t \in (0, t^\star]$, SpecGF outperforms GF with a growing minority-class loss gap $\mathcal{L}_m^{\text{GF}}(t) - \mathcal{L}_m^{\text{Spec}}(t) \ge \mu t / 4$. Moreover, if $p_m \le \frac{k - 2\mu}{2\mu SNR + k SNR + 2k^2}$, then the gap in the balanced loss also grows as $\mathcal{L}_{\text{bal}}^{\text{GF}}(t) - \mathcal{L}_{\text{bal}}^{\text{Spec}}(t) \ge \mu t / 2$.*

*Proof.* The population loss for class $c$ using iterate $\boldsymbol{W}_t$ is written as

$$\begin{aligned}
\mathcal{L}_c(t) &:= \; \tfrac{1}{2} \mathbb{E} \big\| \boldsymbol{y} - \boldsymbol{W}_t \boldsymbol{x} \big\|_2^2 = \tfrac{1}{2} \mathbb{E} \big[ 1 - 2 \boldsymbol{y}^\top \boldsymbol{W}_t \boldsymbol{x} + \boldsymbol{x}^\top \boldsymbol{W}_t^\top \boldsymbol{W}_t \boldsymbol{x} \big]. \\
&= \tfrac{1}{2} \big[ 1 - 2 \boldsymbol{e}_c^\top \boldsymbol{W}_t \boldsymbol{\mu}_c + \big\| \boldsymbol{W}_t \boldsymbol{\mu}_c \big\|_2^2 + \sigma_x^2 \| \boldsymbol{W}_t \|_F^2 \big] \\
&= \tfrac{1}{2} \big\| \boldsymbol{e}_c - \boldsymbol{W}_t \boldsymbol{\mu}_c \big\|_2^2 + \tfrac{1}{2} \sigma_x^2 \| \boldsymbol{W}_t \|_F^2 \\
&= \tfrac{1}{2} \big\| \boldsymbol{e}_c - \boldsymbol{U} \overline{\boldsymbol{W}}_t \boldsymbol{V}^\top \boldsymbol{\mu}_c \big\|_2^2 + \tfrac{1}{2} \sigma_x^2 \| \boldsymbol{U} \overline{\boldsymbol{W}}_t \boldsymbol{V}^\top \|_F^2 \\
&= \tfrac{1}{2} \big\| \boldsymbol{e}_c - \boldsymbol{e}_c \overline{\boldsymbol{W}}_t[c,c] \mu \big\|_2^2 + \tfrac{1}{2} \sigma_x^2 \| \overline{\boldsymbol{W}}_t \|_F^2.
\end{aligned}$$

Define $\alpha_c(t) := \overline{\boldsymbol{W}}_t[c,c]$. Then the per-class loss in terms of the singular values of $\boldsymbol{W}_t$ is written as

$$\mathcal{L}_c(t) = \tfrac{1}{2} (1 - \mu \alpha_c(t))^2 + \tfrac{1}{2} \sigma_x^2 \sum_{c=1}^{K} \alpha_c^2(t). \tag{7}$$

Using Prop. 1 and Lem. 3.1, shows that the singular values of $\boldsymbol{W}^{\text{GF}}(t)$ and $\boldsymbol{W}_t^{\text{SpecGF}}$ evolve as

$$\alpha_c^{\text{GF}}(t) := \overline{\boldsymbol{W}}^{\text{GF}}(t)[c,c] = \alpha_c \left( 1 - \exp\left( -\frac{p_c \mu}{\alpha_c} t \right) \right), \tag{8}$$

$$\alpha_c^{\text{Spec}}(t) := \overline{\boldsymbol{W}}^{\text{SpecGF}}(t)[c,c] = t \, \mathbb{1}\left[ t \le \alpha_c \right] + \alpha_c \mathbb{1}\left[ t > \alpha_c \right], \tag{9}$$

where $\alpha_c := \frac{s_c^{\mathrm{yx}}}{s_c^{\mathrm{xx}}}$ denotes the ratio of the singular values for class $c$, and using Eqs. (5) and (6) from the proof of Lem. 3.1, $\alpha_c = \frac{p_c \mu}{p_c \mu^2 + \sigma_x^2}$.

The time derivatives of $\alpha_c^{\mathrm{GF}}(t)$ and $\alpha_c^{\mathrm{Spec}}(t)$ for $t \le t^*$ are given by

$$\dot\alpha_c^{\mathrm{GF}}(t) := \frac{d\alpha_c^{\mathrm{GF}}(t)}{dt} = p_c \mu\, e^{-(\sigma_x^2 + p_c \mu^2)t}, \qquad \dot\alpha_c^{\mathrm{Spec}}(t) := \frac{d\alpha_c^{\mathrm{Spec}}(t)}{dt} = 1.$$

Define the gaps

$$\Delta\alpha_c(t) := \alpha_c^{\mathrm{Spec}}(t) - \alpha_c^{\mathrm{GF}}(t) \ge 0, \qquad \Delta\dot\alpha_c(t) := \dot\alpha_c^{\mathrm{Spec}}(t) - \dot\alpha_c^{\mathrm{GF}}(t) = 1 - p_c \mu\, e^{-(\sigma_x^2 + p_c \mu^2)t}. \quad (10)$$

Note that $\Delta\dot\alpha_c(t)$ is *increasing* in $t$.

**Minority–class loss gap.** For a fixed $t$, consider the minority loss gap

$$\Delta\mathcal{L}_m(t) := \mathcal{L}_m^{\mathrm{GD}}(t) - \mathcal{L}_m^{\mathrm{Spec}}(t). \quad (11)$$

Using the per-class loss from Eq. (7) and differentiating,

$$\Delta\dot{\mathcal{L}}_m(t) = -\mu \underbrace{\left[\, (1 - \mu\,\alpha_m^{\mathrm{GF}}(t))\,\dot\alpha_m^{\mathrm{GF}}(t) - (1 - \mu\,\alpha_m^{\mathrm{Spec}}(t))\,\dot\alpha_m^{\mathrm{Spec}}(t) \,\right]}_{\text{Term-1}(\Phi)}$$

$$+\ \sigma_x^2 \underbrace{\left[\, \sum_j \alpha_j^{\mathrm{GF}}(t)\dot\alpha_j^{\mathrm{GF}}(t) - \sum_j \alpha_j^{\mathrm{Spec}}(t)\dot\alpha_j^{\mathrm{Spec}}(t) \,\right]}_{\text{Term-2}(\Psi)}, \quad (12)$$

**Term-1 ($\Phi$).** Add–subtract $(1 - \mu\,\alpha_m^{\mathrm{GF}}(t))\dot\alpha_m^{\mathrm{Spec}}(t)$ to Term-1, and we have

$$\begin{aligned}
\Phi &= (1 - \mu\,\alpha_m^{\mathrm{GF}}(t))\dot\alpha_m^{\mathrm{GF}}(t) - (1 - \mu\,\alpha_m^{\mathrm{Spec}}(t))\dot\alpha_m^{\mathrm{Spec}}(t) \\
&= -(1 - \mu\,\alpha_m^{\mathrm{GF}}(t))\,\Delta\dot\alpha_m(t) + \mu\,\Delta\alpha_m(t)\,\dot\alpha_m^{\mathrm{Spec}}(t).
\end{aligned} \quad (13)$$

We know that

$$\Delta\alpha(t) = \int_0^t \Delta\dot\alpha(\tau)\,d\tau \le t\,\Delta\dot\alpha \quad \text{(increasing integrand)}.$$

Substituting in Eq. (13), we get

$$\begin{aligned}
\Phi &\le -\left(1 - \mu\,\alpha_m^{\mathrm{GF}}(t) - t\,\dot\alpha_m^{\mathrm{Spec}}(t)\right)\Delta\dot\alpha_m(t) \\
&\le -\left(1 - \mu\,\alpha_m - t\right)\Delta\dot\alpha_m(t) \\
&\le -\left(1 - 2\mu\,\alpha_m\right)\Delta\dot\alpha_m(t).
\end{aligned} \quad (14)$$

The second and third inequalities assume that $\Delta\dot\alpha_m(t) \ge 0$, which holds true under $p_m \le 1/2\mu$, which is true as we assume $\mu \ge 1$. The third inequality also uses the fact that $t \le t^* = \alpha_m$.

**Term-2 ($\Psi$)** Before any $\alpha_j^{\mathrm{Spec}}$ saturates,

$$\sum_j \alpha_j^{\mathrm{Spec}}\dot\alpha_j^{\mathrm{Spec}} = kt,$$

which gives for all $t \le t^*$ that

$$\Psi \ge -k\alpha_m. \quad (15)$$

$$\begin{aligned}
\Delta\dot{\mathcal{L}}_m(t) &\ge \mu(1 - 2\mu\,\alpha_m)\Delta\dot\alpha_m(t) - \sigma_x^2 k\alpha_m \\
&\ge \mu(1 - 2\mu\,\alpha_m)\,(1 - \mu\,p_m) - \sigma_x^2 k\alpha_m \\
&\ge \tfrac{\mu}{4},
\end{aligned} \quad (16)$$

Here, we first use $p_m \mu \le 1/2$ as $\mu \ge 1$ and $p_m \le 1/2$ as it is miniority class. Next, we use the definition of $\alpha_m = \frac{p_m \mu}{p_m \mu^2 + \sigma_x^2}$ and the assumption $p_m \le \frac{1}{3\mathrm{SNR} + 4k}$. Since $\Delta\mathcal{L}_m(0) = 0$, integrating over $(0, t]$ gives the final bound.

**Class–balanced loss gap.** For a fixed $t$, the class-balanced loss gap is

$$\Delta\mathcal{L}_{\mathrm{bal}}(t) := \sum_c \mathcal{L}_c^{\mathrm{GD}}(t) - \mathcal{L}_c^{\mathrm{Spec}}(t) \tag{17}$$

Using the per-class-loss from Eq. (7) and differentiating,

$$
\begin{aligned}
\Delta\dot{\mathcal{L}}_{\mathrm{bal}}(t) &= -\tfrac{1}{k}\sum_c(1-\mu\alpha_c^{\mathrm{GF}}(t))\mu\dot\alpha_c^{\mathrm{GF}}(t) + \sigma_x^2\sum_c\alpha_c^{\mathrm{GF}}(t)\dot\alpha_c^{\mathrm{GF}}(t)\\
&\quad + \tfrac{1}{k}\sum_c(1-\mu\alpha_c^{\mathrm{Spec}}(t))\mu\dot\alpha_c^{\mathrm{Spec}}(t) - \sigma_x^2\sum_c\alpha_c^{\mathrm{Spec}}(t)\dot\alpha_c^{\mathrm{Spec}}(t)\\
&\geq -\tfrac{1}{k}\sum_c\mu\dot\alpha_c^{\mathrm{GF}}(t) + \tfrac{1}{k}\sum_c(1-\mu\alpha_c^{\mathrm{Spec}}(t))\mu\dot\alpha_c^{\mathrm{Spec}}(t) - \sigma_x^2\sum_c\alpha_c^{\mathrm{Spec}}(t)\dot\alpha_c^{\mathrm{Spec}}(t)\\
&\geq -\tfrac{1}{k}\mu^2 + (1-\mu t)\mu - \sigma_x^2 kt\\
&\geq -\tfrac{1}{k}\mu^2 + \mu - p_m\mu\tfrac{\mathrm{SNR}+k}{p_m\,\mathrm{SNR}+1}.
\end{aligned}
$$

The first inequality follows by using $\alpha_c^{\mathrm{GF}}(t)\dot\alpha_c^{\mathrm{GF}}(t)\geq 0$. For the second inequality, we use $\dot\alpha_c^{\mathrm{GF}}(t)\geq \mu p_c$, and substitute the expressions for $\alpha_c^{\mathrm{Spec}}(t)$ and $\dot\alpha_c^{\mathrm{Spec}}(t)$. In the last step, we use $t\leq\alpha_m$, and the definition of $\mathrm{SNR}$.

As we assume $p_m \leq \frac{k-2\mu}{2\mu\,\mathrm{SNR}+k\,\mathrm{SNR}+2k^2}$, we have that

$$
\begin{aligned}
\Delta\dot{\mathcal{L}}_{\mathrm{bal}}(t) &\geq \mu - \tfrac{1}{k}\mu^2 - p_m\mu\tfrac{\mathrm{SNR}+k}{p_m\,\mathrm{SNR}+1}\\
&\geq \tfrac{\mu}{2}.
\end{aligned}
$$

Integrating this over $(0,t]$, we get $\Delta\mathcal{L}_{\mathrm{bal}}(t)\geq \mu t/2$ for $t\leq t^*$. $\qquad\square$

## B.4 PROOF OF THEOREM 2

We first state the full version of Theorem 2 followed by its proof.

**Theorem 4.** *Assume data model* (DM), *zero initialization, and gradient flow algorithms NGF and SpecGF. Further, assume that $p_M - p_m \geq \frac{2\,p_m\,(p_m\,\mathrm{SNR}+1)^2}{(1-p_m(\mathrm{SNR}+2k))}, p_m < \frac{1}{\mathrm{SNR}+2k}$ Then, for every $t\in(0,t^*]$, SpecGF outperforms NGF with a growing minority-class loss gap $\mathcal{L}_m^{\mathrm{NGF}}(t) - \mathcal{L}_m^{\mathrm{Spec}}(t) \geq \mu t/2$. Moreover, if $p_M - p_m \geq \frac{2(p_m\,\mathrm{SNR}+1)^2}{k(1-p_m(\mathrm{SNR}+2k))}$, then the gap in the balanced loss also grows as $\mathcal{L}_{\mathrm{bal}}^{\mathrm{NGF}}(t) - \mathcal{L}_{\mathrm{bal}}^{\mathrm{Spec}}(t) \geq \mu t/2$.*

*Proof.* The continuous time update of $\alpha_c^{\mathrm{Spec}}(t)$ for $t\leq t^* = \alpha_m$ is given by

$$\dot\alpha_i^{\mathrm{Spec}}(t) = 1, \qquad \alpha_i^{\mathrm{Spec}}(t) = t.$$

In the continuous time limit, the NGF update can be written as

$$\dot\alpha_i^{\mathrm{NGF}}(t) = \frac{r_i(t)}{R(t)}, \qquad r_i(t) := s_i^{\mathrm{yx}} - s_i^{\mathrm{xx}}\alpha_i^{\mathrm{NGF}}(t), \qquad R(t) := \sqrt{\sum_{j=1}^k r_j^2(t)}.$$

We have $0 \leq r_i(t) \leq s_i^{\mathrm{yx}} = \mu p_i$, $R(t) \geq \max_i r_i(t)$, which gives $\dot\alpha_i^{\mathrm{NGF}}(t) \leq 1$ and hence $\alpha_i^{\mathrm{NGF}}(t) \leq t \leq t^* = \alpha_m$. Using this, we get $R(t) \geq s_M^{\mathrm{yx}} - s_M^{\mathrm{xx}}\alpha_m$. Using Lemma 3.1, we have $R(t) \geq p_M\mu - \frac{p_M\mu^2+\sigma_x^2}{p_m\mu^2+\sigma_x^2}p_m\mu = \mu\sigma_x^2\frac{p_M-p_m}{p_m\mu^2+\sigma_x^2}$.

Using Eq. (7), we can write the minority class loss gap as

$$\Delta\mathcal{L}_m(t) := \mathcal{L}_m^{\text{NGD}}(t) - \mathcal{L}_m^{\text{Spec}}(t)$$

$$= \tfrac{1}{2}\left((1 - \mu\alpha_m^{\text{NGF}}(t))^2 + \sigma_x^2\sum_c(\alpha_c^{\text{NGF}}(t))^2 - ((1 - \mu\alpha_m^{\text{Spec}}(t))^2 + \sigma_x^2\sum_c(\alpha_c^{\text{Spec}}(t))^2)\right)$$

$$\implies \Delta\dot{\mathcal{L}}_m(t) = -(1 - \mu\alpha_m^{\text{NGF}}(t))\mu\dot{\alpha}_m^{\text{NGF}}(t) + \sigma_x^2\sum_c\alpha_c^{\text{NGF}}(t)\dot{\alpha}_c^{\text{NGF}}(t)$$

$$+ (1 - \mu\alpha_m^{\text{Spec}}(t))\mu\dot{\alpha}_m^{\text{Spec}}(t) - \sigma_x^2\sum_c\alpha_c^{\text{Spec}}(t)\dot{\alpha}_c^{\text{Spec}}(t)$$

$$\geq -\frac{r_m(t)}{R(t)}\mu + (1 - \mu t)\mu - \sigma_x^2 k t$$

$$\geq \mu - \alpha_m(\mu^2 + k\sigma_x^2) - \frac{p_m(p_m\mu^2 + \sigma_x^2)}{\sigma_x^2(p_M - p_m)}\mu.$$

Since we assume that

$$p_M - p_m \geq \frac{2\,p_m\,(p_m\,\text{SNR}+1)^2}{1 - p_m(\text{SNR}+2k)}, \quad p_m < \frac{1}{\text{SNR}+2k},$$

we have

$$\Delta\dot{\mathcal{L}}_m(t) \geq \mu - \alpha_m(1 + k\sigma_x^2) - \frac{\sigma_x^2 p_m(p_m\mu^2 + \sigma_x^2)(\sigma_x^2 - p_m(\mu^2 + 2k\sigma_x^2))}{2\sigma_x^2\,p_m\,(p_m\mu^2 + \sigma_x^2)^2}\mu$$

$$= \mu - \frac{p_m\mu(\mu^2 + k\sigma_x^2)}{p_m\mu^2 + \sigma_x^2} - \frac{(\sigma_x^2 + p_m\mu^2 - 2p_m(\mu^2 + k\sigma_x^2))}{2(p_m\mu^2 + \sigma_x^2)}\mu$$

$$= \mu/2.$$

Using this, we get $\mathcal{L}_m^{\text{NGD}}(t) - \mathcal{L}_m^{\text{Spec}}(t) \geq \mu t/2$ for $0 \leq t \leq t^*$.

Similarly, for class-balanced loss, we have that

$$\Delta\mathcal{L}_{\text{bal}}(t) := \mathcal{L}_{\text{bal}}^{\text{NGD}}(t) - \mathcal{L}_{\text{bal}}^{\text{Spec}}(t)$$

$$= \tfrac{1}{2k}\sum_c(1 - \mu\alpha_c^{\text{NGF}}(t))^2 + \sigma_x^2\sum_c(\alpha_c^{\text{NGF}}(t))^2 - \left(\tfrac{1}{2k}\sum_c(1 - \mu\alpha_c^{\text{Spec}}(t))^2 + \sigma_x^2\sum_c(\alpha_c^{\text{Spec}}(t))^2\right)$$

$$\Delta\dot{\mathcal{L}}_{\text{bal}}(t) = -\tfrac{1}{k}\sum_c(1 - \mu\alpha_c^{\text{NGF}}(t))\mu\dot{\alpha}_c^{\text{NGF}}(t) + \sigma_x^2\sum_c\alpha_c^{\text{NGF}}(t)\dot{\alpha}_c^{\text{NGF}}(t)$$

$$+ \tfrac{1}{k}\sum_c(1 - \mu\alpha_c^{\text{Spec}}(t))\mu\dot{\alpha}_c^{\text{Spec}}(t) - \sigma_x^2\sum_c\alpha_c^{\text{Spec}}(t)\dot{\alpha}_c^{\text{Spec}}(t)$$

$$\geq -\tfrac{1}{k}\sum_c\frac{r_c(t)}{R(t)}\mu + (1 - \mu t)\mu - \sigma_x^2 k t$$

$$\geq \mu - \alpha_m(\mu^2 + k\sigma_x^2) - \frac{(p_m\mu^2 + \sigma_x^2)}{k\sigma_x^2(p_M - p_m)}\mu,$$

where we use $\sum_c r_c(t) \leq \mu\sum_c p_c = \mu$. Since we assume that

$$p_M - p_m \geq \frac{2(p_m\,\text{SNR}+1)^2}{k(1 - p_m(\text{SNR}+2k))}, \quad p_m < \frac{1}{\text{SNR}+2k},$$

we have

$$\Delta\dot{\mathcal{L}}_{\text{bal}}(t) \geq \mu - \alpha_m(\mu^2 + k\sigma_x^2) - \frac{\sigma_x^2\left(\sigma_x^2(1 - 2kp_m) - p_m\mu^2\right)(p_m\mu^2 + \sigma_x^2)}{2\sigma_x^2\left(p_m\mu^2 + \sigma_x^2\right)^2}\mu$$

$$= \mu\left(1 - \frac{p_m(\mu^2 + k\sigma_x^2)}{p_m\mu^2 + \sigma_x^2} - \frac{2\sigma_x^2\left(1 - kp_m\right) - (p_m\mu^2 + \sigma_x^2)}{2(p_m\mu^2 + \sigma_x^2)}\right)$$

$$= \mu/2.$$

Using this, we get $\mathcal{L}_{\text{bal}}^{\text{NGD}}(t) - \mathcal{L}_{\text{bal}}^{\text{Spec}}(t) \geq \mu t/2$ for $0 \leq t \leq t^*$. $\qquad\square$

## B.5 PROOF OF PROPOSITION 2

We first state the full version of Proposition 2 followed by its proof.

**Proposition 3.** *Suppose Condition 1 holds. Let the weights be initialized as $W_0^0 = e^{-\delta}Q[I_{d_1}\ \mathbf{0}_{d-d_1}]V^\top$ and $W_0^1 = e^{-\delta}U[I_k\ \mathbf{0}_{d_1-k}]Q^\top$, where $Q \in \mathbb{R}^{d_1 \times d_1}$ is an orthonormal matrix, $k < d_1 < d$, and $\delta > 0$ is a constant. Then, for SpecGD, at each iteration, $W_t^0 := Q\overline{W}_t^0 V^\top$, $W_t^1 = U\overline{W}_t^1 Q^\top$, where $\overline{W}_t^0, \overline{W}_t^1$ are zero except their main diagonal along which the entries evolve as follows for $i \in [k]$:*

$$\overline{W}_t^0[i,i] = \overline{W}_t^1[i,i] = (\eta\, t + e^{-\delta})\, \mathbb{1}\left[t \leq \frac{1}{\eta}\left(\sqrt{\frac{s_i^{yx}}{s_i^{xx}}} - e^{-\delta}\right)\right] + \sqrt{\frac{s_i^{yx}}{s_i^{xx}}}\, \mathbb{1}\left[t > \frac{1}{\eta}\left(\sqrt{\frac{s_i^{yx}}{s_i^{xx}}} - e^{-\delta}\right)\right].$$

*Proof.* Here, we assume that $d > d_1 > k$.[†] We can write the gradients as

$$\nabla_{W^0}\mathcal{L}(W_t^0, W_t^1) = -(W_t^1)^\top \mathbb{E}\left[(y - W_t^1 W_t^0 x)x^\top\right] = -(W_t^1)^\top \Sigma_{yx} + (W_t^1)^\top W_t^1 W_t^0 \Sigma_x,$$

$$\nabla_{W^1}\mathcal{L}(W_t^0, W_t^1) = -\mathbb{E}\left[(y - W_t^1 W_t^0 x)x^\top (W_t^0)^\top\right] = -\Sigma_{yx}(W_t^0)^\top + W_t^1 W_t^0 \Sigma_x (W_t^0)^\top.$$

Let $\overline{W}_t^0 := Q^\top W_t^0 V$ and $\overline{W}_t^1 := U^\top W_t^1 Q$. Define $\overline{W}_t = \overline{W}_t^1 \overline{W}_t^0$. Then, we can write

$$\nabla_{W^0}\mathcal{L}(W_t^0, W_t^1) = -Q(\overline{W}_t^1)^\top S_{yx} V^\top + Q(\overline{W}_t^1)^\top \overline{W}_t S_{xx} V^\top,$$

$$\nabla_{W^1}\mathcal{L}(W_t^0, W_t^1) = -U S_{yx}(\overline{W}_t^0)^\top Q^\top + U\overline{W}_t S_{xx}(\overline{W}_t^0)^\top Q^\top.$$

This follows by writing $W_t^0 = Q\overline{W}_t^0 V^\top$, $W_t^1 = U\overline{W}_t^1 Q^\top$ and substituting in the gradient expressions.

We now simplify the gradients at the first iteration. Note that we have

$$\overline{W}_0^0 = e^{-\delta}[I_{d_1}\ \mathbf{0}_{d-d_1}], \quad \overline{W}_0^1 = e^{-\delta}[I_k\ \mathbf{0}_{d_1-k}],$$

$$\overline{W}_0 = e^{-2\delta}[I_k\ \mathbf{0}_{d-k}].$$

Using these, we can get gradients at first step as

$$\nabla_{W^0}\mathcal{L}(W_0^0, W_0^1) = -e^{-\delta}Q[I_k\ \mathbf{0}_{d_1-k}]^\top\left(S_{yx} - e^{-2\delta}[I_k\ \mathbf{0}_{d-k}]S_{xx}\right)V^\top,$$

$$\nabla_{W^1}\mathcal{L}(W_0^0, W_0^1) = -e^{-\delta}U\left(S_{yx} - e^{-2\delta}[I_k\ \mathbf{0}_{d-k}]S_{xx}\right)[I_k\ \mathbf{0}_{d-d_1}]^\top Q^\top.$$

Then, for sufficiently small initialization, the iterates after one step of SpecGD are written as

$$W_1^0 = e^{-\delta}Q[I_{d_1}\ \mathbf{0}_{d-d_1}]V^\top + \eta Q[I_k\ \mathbf{0}_{d_1-k}]^\top (V[I_k\ \mathbf{0}_{d-k}]^\top)^\top,$$

$$W_1^1 = e^{-\delta}U[I_k\ \mathbf{0}_{d_1-k}]Q^\top + \eta U(Q[I_k\ \mathbf{0}_{d_1-k}]^\top)^\top.$$

Using this iterate after step 1, we can see that $\overline{W}_1^0, \overline{W}_1^1, \overline{W}_1$ remain diagonal. Using the gradient expressions, we see that the update at step 1 remains in the same basis. This implies the update remains in the same basis. Proceeding similarly, we can see that with the given initialization, $\overline{W}_t^0$ and $\overline{W}_t^1$ remain diagonal, and the updates for $\overline{W}_t^0$ and $\overline{W}_t^1$ using SpecGD can be written as

$$\overline{W}_{t+1}^0 = \overline{W}_t^0 + \eta\left(\sum_{i:\,\overline{W}_t[i,i]\,s_i^{xx} < s_i^{yx}} e_i e_i^\top\right)\begin{bmatrix} I_k & \mathbf{0}_{d-k} \\ \mathbf{0}_{d_1-k} & \mathbf{0}_{d-k} \end{bmatrix},$$

$$\overline{W}_{t+1}^1 = \overline{W}_t^1 + \eta \sum_{i:\,\overline{W}_t[i,i]\,s_i^{xx} < s_i^{yx}} e_i e_i^\top [I_k\ \mathbf{0}_{d_1-k}].$$

Simplifying, we get

$$\overline{W}_t^0[i,i] = \overline{W}_t^1[i,i] = (\eta\, t + e^{-\delta})\, \mathbb{1}\left[t \leq \frac{1}{\eta}\left(\sqrt{\frac{s_i^{yx}}{s_i^{xx}}} - e^{-\delta}\right)\right] + \sqrt{\frac{s_i^{yx}}{s_i^{xx}}}\, \mathbb{1}\left[t > \frac{1}{\eta}\left(\sqrt{\frac{s_i^{yx}}{s_i^{xx}}} - e^{-\delta}\right)\right].$$

$\square$

---

[†] The proof works for other cases as well and we just use this one for the purpose of illustrating the technique.

## B.6 SpecGD Dynamics for Deep Linear Model

The following result characterizes the dynamics of SpecGD for a deep $L$-layer linear model ($L \geq 2$).

**Proposition 4.** *Assume $k = d$, $L \geq 2$, and suppose Ass. 1 holds. Let weights be initialized as*

$$\boldsymbol{W}_0^0 = e^{-\delta}\boldsymbol{Q}_1\boldsymbol{V}^\top, \quad \boldsymbol{W}_0^l = e^{-\delta}\boldsymbol{Q}_{l+1}\boldsymbol{Q}_l^\top, \quad \boldsymbol{W}_0^{L-1} = e^{-\delta}\boldsymbol{U}\boldsymbol{Q}_{L-1}^\top,$$

*where $\{\boldsymbol{Q}_l\}_{l=1}^{L-1}$ are orthonormal matrices and $\delta > 0$ is a constant. Then, for SpecGD, at each iteration, $\boldsymbol{W}_t^0 := \boldsymbol{Q}_1\overline{\boldsymbol{W}}_t^0\boldsymbol{V}^\top$, $\boldsymbol{W}_t^l := \boldsymbol{Q}_{l+1}\overline{\boldsymbol{W}}_t^l\boldsymbol{Q}_l^\top$, $\boldsymbol{W}_t^{L-1} = \boldsymbol{U}\overline{\boldsymbol{W}}_t^{L-1}\boldsymbol{Q}_{L-1}^\top$, where $\overline{\boldsymbol{W}}_t^l$ for $l \in [L]$ is zero except its main diagonal along which entries evolve as follows for $i \in [k]$:*

$$\overline{W}_t^l[i,i] = (\eta t + e^{-\delta})\,\mathbb{1}\left[t \leq \frac{1}{\eta}\left(\left(\frac{s_i^{yx}}{s_i^{xx}}\right)^{1/L} - e^{-\delta}\right)\right] + \left(\frac{s_i^{yx}}{s_i^{xx}}\right)^{1/L}\mathbb{1}\left[t > \frac{1}{\eta}\left(\left(\frac{s_i^{yx}}{s_i^{xx}}\right)^{1/L} - e^{-\delta}\right)\right].$$

*Further, the gap between saturation of the minority vs. majority component is $\Delta T := \frac{t_{\max} - t_{\min}}{t_{\min}} = \left(\frac{s_1^{yx}/s_1^{xx}}{s_k^{yx}/s_k^{xx}}\right)^{1/L} - 1.$*

*Proof.* We have that $\overline{\boldsymbol{W}}^0 = (\boldsymbol{Q}_1)^\top\boldsymbol{W}^0\boldsymbol{V}$, $\overline{\boldsymbol{W}}^l = (\boldsymbol{Q}_{l+1})^\top\boldsymbol{W}^l\boldsymbol{Q}_l$ and $\overline{\boldsymbol{W}}^{L-1} = \boldsymbol{U}^\top\boldsymbol{W}^{L-1}\boldsymbol{Q}_{L-1}$, where $l \in \{1, \ldots, L-2\}$. Also define

$$\boldsymbol{W} = \prod_{i=0}^{L-1}\boldsymbol{W}^{L-1-i}, \quad \boldsymbol{W}^{>l} = \prod_{i=l+1}^{L-1}(\boldsymbol{W}^i)^\top, \quad \boldsymbol{W}^{<l} = \prod_{i=0}^{l-1}(\boldsymbol{W}^i)^\top.$$

Then, we can write the gradients as

$$\nabla_{\boldsymbol{W}^l}\mathcal{L}(\boldsymbol{W}_t^0, \ldots, \boldsymbol{W}_t^{L-1}) = -\boldsymbol{W}_t^{>l}\mathbb{E}\left[(\boldsymbol{y} - \boldsymbol{W}_t\boldsymbol{x})\boldsymbol{x}^\top\right]\boldsymbol{W}_t^{<l} = -\boldsymbol{W}_t^{>l}\boldsymbol{\Sigma}_{\boldsymbol{yx}}\boldsymbol{W}_t^{<l} + \boldsymbol{W}_t^{>l}\boldsymbol{W}_t\boldsymbol{\Sigma}_{\boldsymbol{x}}\boldsymbol{W}_t^{<l},$$

Note that through change of basis, we can write

$$\boldsymbol{W} = \boldsymbol{U}\prod_{i=0}^{L-1}\overline{\boldsymbol{W}}^{L-1-i}\boldsymbol{V}^\top, \quad \boldsymbol{W}^{>l} = \boldsymbol{Q}_{l+1}\prod_{i=l+1}^{L-1}(\overline{\boldsymbol{W}}^i)^\top\boldsymbol{U}^\top, \quad \boldsymbol{W}^{<l} = \boldsymbol{V}\prod_{i=0}^{l-1}(\overline{\boldsymbol{W}}^i)^\top\boldsymbol{Q}_l^\top,$$

and consequently, for $1 \leq l \leq L-2$,

$$\nabla_{\boldsymbol{W}^l}\mathcal{L}(\boldsymbol{W}_t^0, \ldots, \boldsymbol{W}_t^{L-1}) = -\boldsymbol{Q}_{l+1}\prod_{i=l+1}^{L-1}(\overline{\boldsymbol{W}}_t^i)^\top\left(\boldsymbol{S}_{\boldsymbol{yx}} - \prod_{i=0}^{L-1}\overline{\boldsymbol{W}}_t^{L-1-i}\boldsymbol{S}_{\boldsymbol{xx}}\right)\prod_{i=0}^{l-1}(\overline{\boldsymbol{W}}_t^i)^\top\boldsymbol{Q}_l^\top.$$

We can see that $\overline{\boldsymbol{W}}_0^l$ is diagonal at initialization. Then, if for $j \in [d]$, $s_j^{yx} > \prod_{i=0}^{L-1}\overline{W}_0^i[j,j]s_j^{xx}$, the SpecGD update is

$$\boldsymbol{W}_1^l = \boldsymbol{W}_0^l + \eta\boldsymbol{Q}_{l+1}\boldsymbol{Q}_l^\top.$$

Proceeding similarly, for any $l \in [L-1]$, we can write the SpecGD updates at any $t > 0$ as

$$\overline{\boldsymbol{W}}_{t+1}^l = \overline{\boldsymbol{W}}_t^l + \eta\sum_{j:\prod_{i=0}^{L-1}\overline{W}_t^i[j,j]s_j^{xx}<s_j^{yx}}\boldsymbol{e}_j\boldsymbol{e}_j^\top.$$

Simplifying, we get

$$\overline{W}_t^l[i,i] = (\eta t + e^{-\delta})\,\mathbb{1}\left[t \leq \frac{1}{\eta}\left(\left(\frac{s_i^{yx}}{s_i^{xx}}\right)^{1/L} - e^{-\delta}\right)\right] + \left(\frac{s_i^{yx}}{s_i^{xx}}\right)^{1/L}\mathbb{1}\left[t > \frac{1}{\eta}\left(\left(\frac{s_i^{yx}}{s_i^{xx}}\right)^{1/L} - e^{-\delta}\right)\right].$$

$\square$

## C    ADDITIONAL RESULTS IN THE LINEAR SETTING

In this section, we present some additional experimental results for the synthetic setting with class imbalance considered in Sec. 3, where we use a linear model and MSE loss. We consider the same setting and parameter values as in Fig. 2, unless stated otherwise.

In Fig. 10, we compare the dynamics of SignGD, Adam (with $\beta_1 = 0.9$, $\beta_2 = 0.999$) and Muon (with $\beta = 0.9$) with GD, NGD and SpecGD. We find (from the rightmost plot) that in contrast to (N)GD and SpecGD iterates that remain in the $\boldsymbol{UV}^\top$ eigenbasis (defined as in Condition 1), SignGD and Adam iterates take some time to align to that basis. We also find that SignGD and Adam learn the spectral components in a more balanced way compared to (N)GD, and get slightly better worst-class loss early on in training. However, since SpecGD learns the spectral components at an equal rate, it attains the best worst-class and class-balanced loss early on. Notably, the dynamics of Muon closely follow those of SpecGD. All optimizers eventually lead to convergence to the same solution.

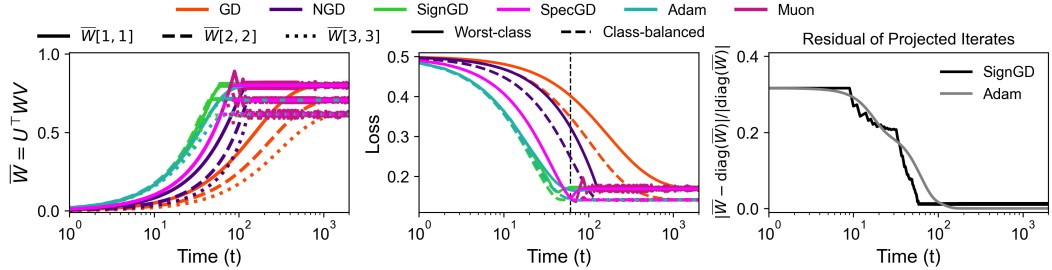

Figure 10: Dynamics of the iterates (left) and loss (middle) for GD, NGD, SignGD, SpecGD, Adam and Muon on a linear model with class imbalance (priors $p_1 > p_2 > p_3$). SpecGD and Muon learn all spectral components at the same rate, whereas (N)GD prioritizes more dominant components. SignGD and Adam also seem to learn the spectral components at the same rate; however, their iterates take time to align to the $\boldsymbol{UV}^\top$ eigenbasis (as seen from the residual of projected iterates (right)). Although all converge to the same solution, SpecGD and Muon significantly outperform in worst-class and class-balanced loss early on in training.

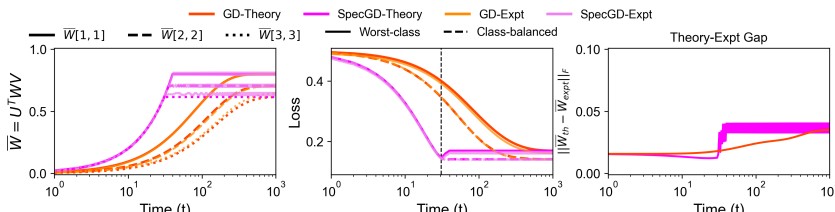

Figure 11: Comparison of the dynamics of iterates (left) and loss (middle) for GD and SpecGD on a linear model with class imbalance (priors $p_1 > p_2 > p_3$) in the population limit with zero initialization (theory) and in the finite sample setting with random initialization (expt), and norm of the theoretical/experimental difference between the iterates (right). We find that the dynamics characterizes by the theory closely match the experimental behaviour.

In Fig. 11, we compare the dynamics of GD and SpecGD under the theoretical setting (population limit and zero initialization) vs. the experimental setting (using 2000 samples with zero-mean Gaussian initialization with variance $10^{-2}$). We use a learning rate of 0.02 in this case. For comparing the iterates, we project them onto the $\boldsymbol{UV}^\top$ eigenbasis from the population setting. We observe that the iterate and loss dynamics from theory closely predict the behaviour seen in the experimental setting.

In Fig. 12, we consider the same theoretical and experimental setting as in Fig. 11, and compare the dynamics of SpecGD in both cases to the dynamics of Muon (with $\beta = 0.9$) in the experimental setting. We use a learning rate of 0.005 in this case. We observe that the iterate and loss dynamics for Muon closely follow those predicted theoretically for SpecGD.

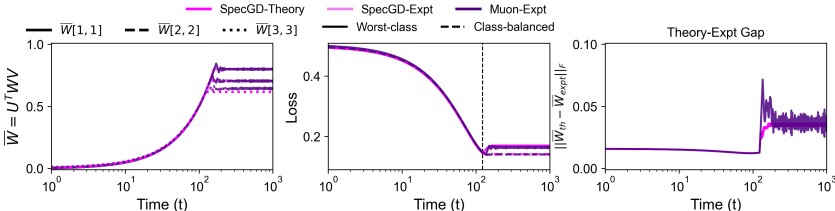

Figure 12: Comparison of the dynamics of iterates (left) and loss (middle) for SpecGD (both in the population limit with zero initialization (theory) and in the finite sample setting with random initialization (expt)) and Muon (expt) on a linear model with class imbalance (priors $p_1 > p_2 > p_3$), and norm of the difference between the iterates (SpecGD-theory vs SpecGD/Muon expt) (right). We find that the dynamics characterizes by the theory closely match the experimental behaviour.

In the next experiment, we examine the effect of relaxing Condition 1 on the dynamics of GD and SpecGD iterates. Specifically, in the relaxed version, the joint diagonalization holds with $S_{xx} + B$, instead of just a diagonal matrix $S_{xx}$, such that $\|B\| \leq \epsilon$. We note that while Condition 1 is satisfied for our data model (DM) in the population limit, only the relaxed version holds in the finite sample setting. For instance, under the setting considered in Fig. 11, $\|B\| \approx 0.013$. In Fig. 13, we compare the dynamics of iterates in the finite sample setting ($\|B\| = \epsilon$) to a baseline, where we use the $UV^\top$ eigenbasis from this setting but set $B = 0$. We find that the iterates in the two settings closely follow each other.

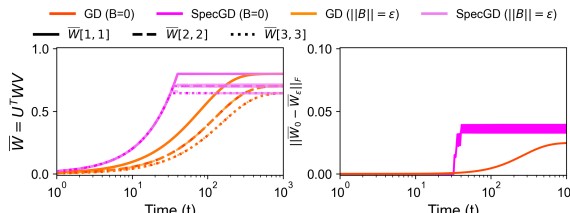

Figure 13: Comparison of the dynamics of iterates (left) for GD and SpecGD on a linear model with class imbalance (priors $p_1 > p_2 > p_3$) when $B = 0$ (Condition 1 holds) vs. when $\|B\| = \epsilon$ (Condition 1 relaxed), and norm of the difference between the iterates (right). We find that the dynamics in the latter setting closely match those under Condition 1.

## D  ADDITIONAL RESULTS AND DETAILS OF EXPERIMENTAL SETTINGS

This section presents the details of experimental settings for our experiments with the Colored-MNIST dataset and those in Sec. 2 and Sec. 4, as well as additional results.

For our experiments with Shampoo, we rely on the `distributed-shampoo` reference implementation of Shi et al. (2023) and use matrix inverse stabilization parameter $\varepsilon = 10^{-6}$, `precondition_frequency = 5`, `max_preconditioner_dim = 8192`, unless stated otherwise. To stabilize training, we also use `SGDGraftingConfig` or `AdamGraftingConfig` for the update grafting. Here, $\beta_1$ governs the exponential moving average of the raw gradients, while $\beta_2$ governs the accumulation in the matrix preconditioners (see App. A). Further, $\varepsilon$ is a small diagonal 'jitter' that stabilises the matrix inverse, and SGD/Adam grafting simply rescales the Shampoo update so its overall magnitude matches that of a plain SGD/Adam step (with default $\beta$ parameters).

For our experiments with Muon, we use the implementation from Jordan et al. (2024). Here, $\beta$ represents the momentum parameter, and we perform one Newton-Schulz iteration in each step. By default, the Adam optimizer is used to update all vector parameters. For hyperparameter tuning, we set $\beta_1$ for Adam to be the same as $\beta$ for Muon. We use the default value of $\beta_2$ for Adam with training with Muon.

For our experiments with Adam and SGD, we use the standard implementations from the PyTorch library Paszke et al. (2019). For SGD $\beta$ refers to the momentum parameter whereas for Adam $\beta_1$ refers to momentum and $\beta_2$ is the exponential moving average parameter (see App. A).

Following recent works (Srećković et al., 2025; Marek et al., 2025), which show that the performance gap between SGD and adaptive optimizers like Adam is larger for larger batch sizes, we use larger batch sizes in our experiments.

### D.1 COLORED-MNIST

*Dataset.* In Sec. 1, we consider a variant of the Colored-MNIST dataset, a benchmark used commonly in the literature on spurious correlations (Arjovsky et al., 2020; Pezeshki et al., 2021). The task is to classify each digit as either $< 5$ or $\geq 5$. The digits in each class are injected with a background color (red or green) that is correlated with the label and acts as a spurious feature. In our setting, the spurious correlation in the train set is $99\%$.

*Model Architecture and Training.* We train a four-layer ReLU MLP with hidden layer dimensions 512, 128, and 32, using inputs from the standard Coloured-MNIST dataset. Each image is $28 \times 28$ with 3 channels (RGB). Training is performed for $500$ epochs in the full-batch setting. We don't use weight decay or a learning rate scheduler in this setting. We use preconditioner frequency 1 for Shampoo.

*Results.* Fig. 14 shows the set of hyperparameters we tried for different optimizers and effect on majority- and minority-group test accuracy. All results are averaged over five independent runs. We select the best hyperparameter setting based on the best average group-balanced accuracy at the end of training.

Building on our initial comparison of optimizers from the Frobenius, max, and spectral norm families in Fig. 1, we present a broader comparison that includes additional optimizers in Fig. 15.

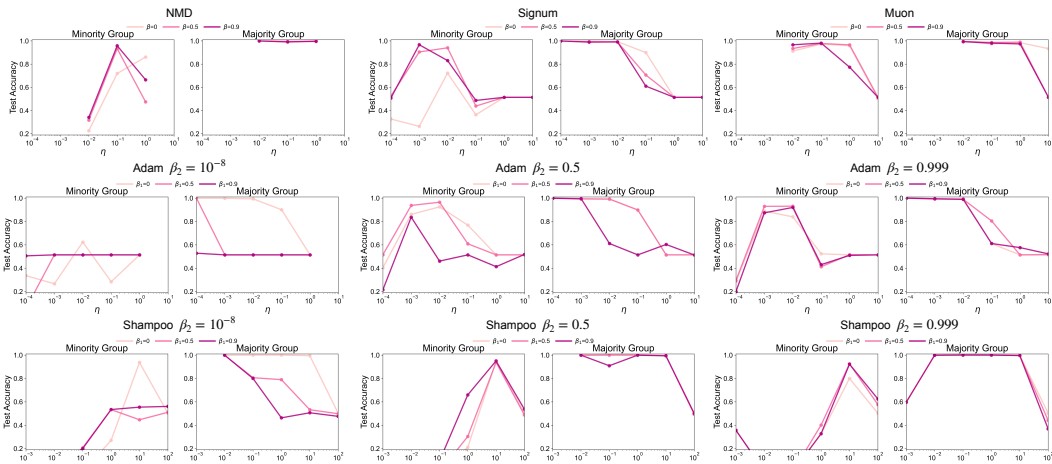

Figure 14: Hyperparameter sweeps for different optimizers on the Colored-MNIST dataset, showing the majority- and minority-group test accuracies at the end of training.

### D.2 CIFAR-10/100 WITH CLASS IMBALANCE

*Dataset and Preprocessing.* We construct class-imbalanced versions of CIFAR-10 and CIFAR-100 (Krizhevsky et al., 2009) using step imbalance with ratio $R = 20$. For CIFAR-10, we sample half the classes (5) to be majority classes containing $20\times$ more samples (5000) than the rest of the (minority) classes (250). The same procedure is applied to CIFAR-100 using all 100 classes.

We apply standard data augmentation including random horizontal flips, random crops, and normalization using dataset-specific channel means and standard deviations. The datasets are split into training and test sets following the original CIFAR splits.

*Model Architecture and Training.* We train ResNet-18 on CIFAR-10 and ResNet-50 on CIFAR-100 following standard architectures (He et al., 2016). All models use random initialization.

Training is conducted with batch size 2048 for 200 epochs. We compare four optimizers representing different norm-based approaches, namely NMD, Signum and Adam, and Muon. We use weight decay 0 for all optimizers. All optimizers use cosine learning rate scheduler. We select the learning rate and momentum by a grid search (see Fig 16, Fig 17). The hyperparameters with the highest minority-class accuracy are used to report the final results. We use the following learning rate ($\eta$) and momentum ($\beta$ or $\beta_1$) configurations:

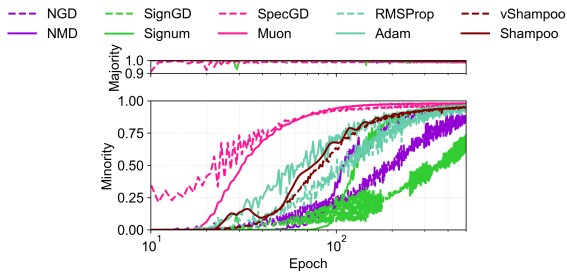

Figure 15: Comparison of optimizer test accuracy on the Colored-MNIST dataset. The top and bottom panels show performance on the majority and minority groups, respectively. We compare several families of optimizers, starting with *Normalized Steepest Descent* methods—NGD, SignGD, and SpecGD—which normalize the gradient using the Frobenius, $\max$, and spectral norms, respectively. We then show their corresponding *Normalized Momentum Descent* variants: NMD, Signum, and Muon. Finally, we compare other adaptive methods, contrasting Adam (element-wise preconditioning) with Shampoo (full-matrix preconditioning), along with their non-momentum analogs, RMSProp and vShampoo, to isolate the effect of preconditioning. We observe that Muon and SpecGD significantly outperform their Frobenius (NMD, NGD) and $\max$ (SignGD, Signum) counterparts in terms of test accuracy on minority groups. However, we do not observe a discernible performance gap when comparing Shampoo with Adam or vShampoo with RMSProp.

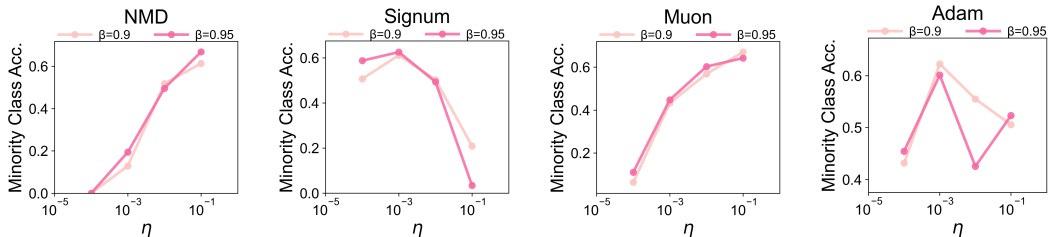

Figure 16: Hyperparameter sweeps for different optimizers on the CIFAR-10 dataset, showing minority-class test accuracies at the end of training.

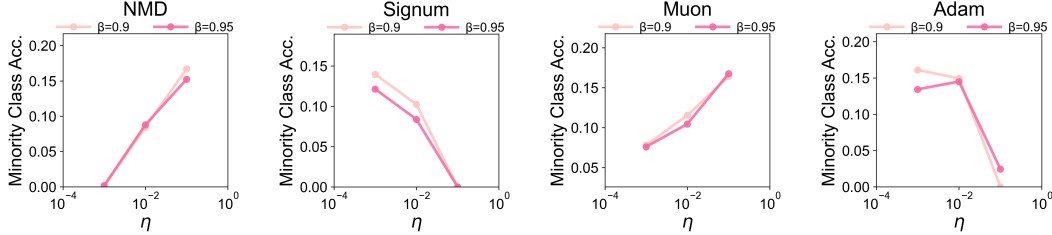

Figure 17: Hyperparameter sweeps for different optimizers on the CIFAR-100 dataset, showing minority-class test accuracies at the end of training.

*CIFAR-10:* NMD ($\eta = 0.1$, $\beta = 0.95$), Signum ($\eta = 0.001$, $\beta = 0.95$), Adam ($\eta = 0.001$, $\beta_1 = 0.9$), Muon ($\eta = 0.1$, $\beta = 0.9$).

*CIFAR-100:* NMD ($\eta = 0.1$, $\beta = 0.9$), Signum ($\eta = 0.001$, $\beta = 0.9$), Adam ($\eta = 0.001$, $\beta_1 = 0.9$), Muon ($\eta = 0.1$, $\beta = 0.95$).

*Results.* As shown in Fig. 18 (CIFAR-10) and Fig. 19 (CIFAR-100), Muon consistently outperforms other optimizers on both majority- and minority-class accuracies for both experiments, with the performance advantage being most pronounced during the early stages of training.

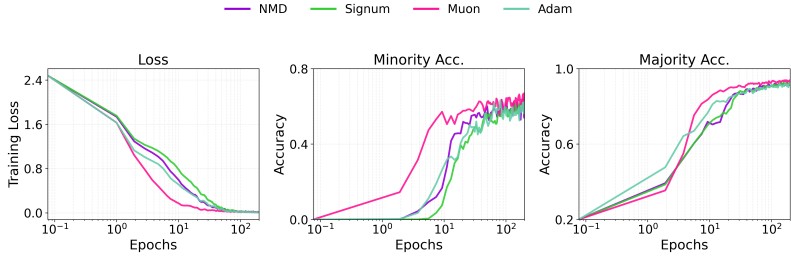

Figure 18: Comparison of training loss, and minority- and majority-class test accuracies when training a ResNet-18 model on CIFAR-10 using different optimizers. Muon achieves superior minority-class test accuracy while maintaining competitive majority-class performance, especially early on in training.

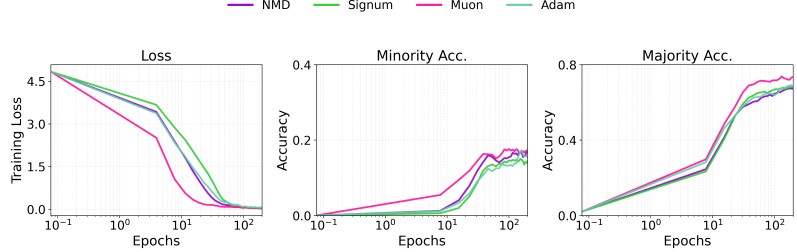

Figure 19: Comparison of training loss, and minority- and majority-class test accuracies when training a ResNet-50 model on CIFAR-100 using different optimizers. Muon achieves superior minority-class test accuracy while maintaining competitive majority-class performance, especially early on in training.

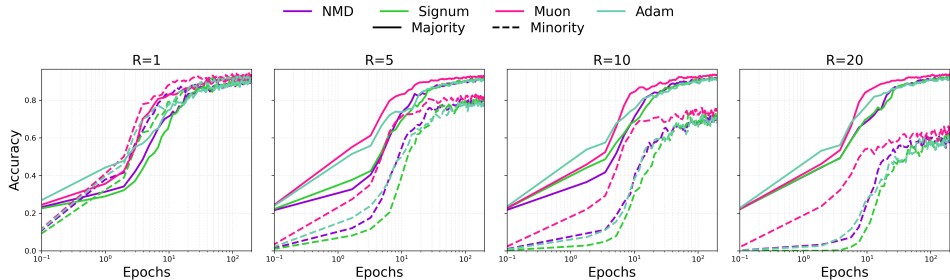

Figure 20: Test accuracy dynamics on CIFAR-10 with ResNet-18 across varying imbalance ratios $R$. Muon consistently outperforms other optimizers on minority classes, with the performance gap becoming more pronounced as $R$ increases.

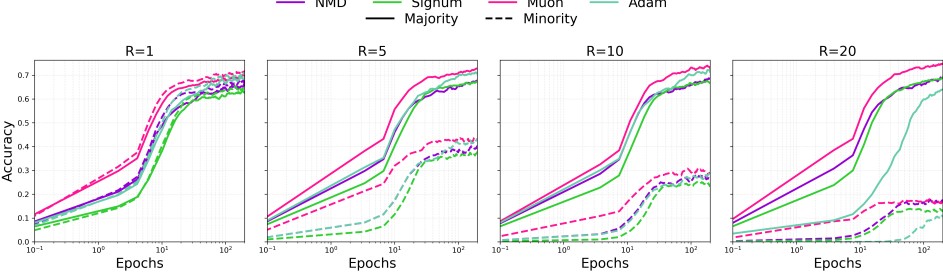

Figure 21: Test accuracy dynamics on CIFAR-100 with ResNet-50 across varying imbalance ratios $R$. In contrast to the behaviour on CIFAR-10, we find that in this setting Muon maintains superior or comparable performance across all imbalance levels.

### D.2.1 EFFECT OF CLASS IMBALANCE RATIO

To investigate how the severity of class imbalance affects the relative performance of different optimizers, we conduct experiments varying the imbalance ratio $R \in \{1, 5, 10, 20\}$ on both CIFAR-10 and CIFAR-100.

*Experimental Setup.* We use the same step-imbalanced construction as in Sec. 2, but vary the imbalance ratio $R$, which denotes the majority-to-minority sample ratio. $R = 1$ corresponds to a balanced dataset, while higher values indicate progressively more severe imbalance. For CIFAR-10, we train ResNet-18, and for CIFAR-100, we train ResNet-50, following the same data augmentation and preprocessing pipeline described in App. D.2.

We use the optimal hyperparameters identified in our $R = 20$ experiments (App. D.2) for each optimizer across all imbalance ratios to ensure fair comparison. All models are trained for 200 epochs with batch size 2048 and cosine learning rate scheduling.

*Results.* Figures 20 and 21 show test accuracy dynamics across varying imbalance ratios. On CIFAR-10, Muon's advantage grows with imbalance severity. This gap is particularly pronounced early in training: while all optimizers perform similarly on balanced data ($R = 1$), at $R = 20$ Muon achieves about $20\%$ minority-class accuracy compared to $< 10\%$ for NMD and Signum at the end of epoch 1. On CIFAR-100, the effect of changing $R$ is less pronounced and Muon maintains the best performance across all settings.

### D.2.2 FINE-TUNING FROM IMAGENET PRETRAINING

In this section, we investigate whether the observations for (pre)training the model from random initialization are consistent with the case where we fine-tune an ImageNet-pretrained model.

*Experimental Setup.* We fine-tune a ResNet-18 model initialized with ImageNet-trained weights (IMAGENET1K_V1) on CIFAR-10 with step imbalance ($R = 20$). While retaining the pretrained weights, we modify three components for CIFAR-10: the first convolutional layer is replaced with a new layer adapted for $32 \times 32$ inputs (kernel size 3, stride 1, padding 1), the maxpooling layer is removed (replaced with identity), and the final fully connected layer is replaced for 10-class output. All layers remain trainable during fine-tuning. Models are fine-tuned for 200 epochs with batch size 2048 and cosine learning rate scheduling.

Following the protocol specified earlier, the hyperparameters used to report the final results are selected based on the best minority-class accuracy. We compare NMD ($\eta = 10^{-2}$, $\beta = 0.99$), Signum ($\eta = 10^{-4}$, $\beta = 0.9$), Adam ($\eta = 10^{-3}$, $\beta_1 = 0.9$, $\beta_2 = 0.999$), and Muon ($\eta = 10^{-2}$, $\beta = 0.9$). All optimizers use zero weight decay.

*Results.* As shown in Fig. 22, when fine-tuning from ImageNet initialization, the optimizer dynamics differ from training from scratch (Fig. 18). Adam, Muon, and Signum achieve comparable training loss and majority-class accuracy, while NMD exhibits slower convergence. For minority-class accuracy, Adam demonstrates the best performance, followed by Muon and Signum, while NMD struggles to learn minority classes quickly.

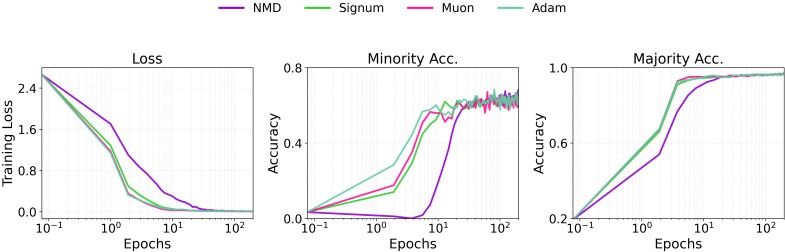

Figure 22: Fine-tuning ImageNet-pretrained ResNet-18 on imbalanced CIFAR-10 ($R = 20$). While all optimizers except NMD achieve similar majority-class accuracy, Adam demonstrates superior minority-class performance, followed by Muon and Signum. NMD shows significantly delayed learning on both majority and minority classes.

### D.3 MNIST-CIFAR DATASET

We train the model for 500 epochs or until convergence (loss reaches 0.01). We use a batch size of 2048 and a cosine annealing learning rate scheduler. We use a fixed weight decay of $10^{-5}$.

For SGD and Muon, we test learning rates $0.01, 0.1$, and momentum values $10^{-8}, 0.5, 0.9$. For Adam and Shampoo, we test learning rates $10^{-4}, 10^{-3}$, $\beta_1$ values $10^{-8}, 0.5, 0.9$ and $\beta_2$ values $10^{-8}, 0.5, 0.999$. The results are averaged over three independent runs. The results are reported for the best hyperparameter setting for each optimizer, selected based on the best seed-averaged worst-group accuracy on the validation set.

### D.4 SUBGROUP ROBUSTNESS BENCHMARKS

For MultiNLI, we use a batch size of $512$ and the results are averaged over four independent runs. For CelebA, we use a batch size of $1024$ and the results are averaged over five independent runs.

To ensure our comparisons are robust, we conduct extensive hyperparameter sweeps for all optimizers, as shown for the MultiNLI dataset in Fig. 23 and CelebA in Fig. 24. This selection protocol confirms that the consistent superior performance of spectral optimizers like Muon and Shampoo over SGD is not an artifact of a specific hyperparameter choice.

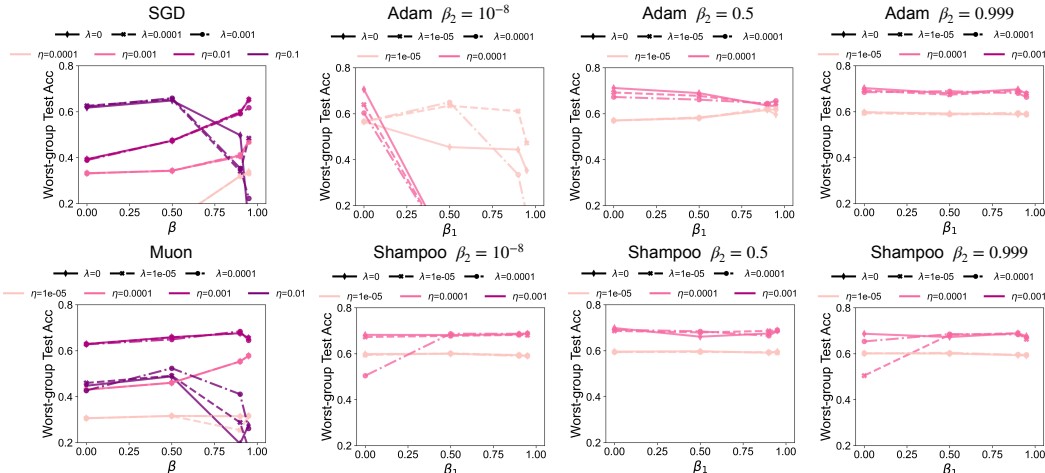

Figure 23: Hyperparameter sweep for different optimizers on the MultiNLI dataset, showing worst-group test accuracy at the last fine-tuning epoch. The sweep varies the momentum parameters ($\beta$ for SGD/Muon, $\beta_1, \beta_2$ for Adam/Shampoo), learning rate ($\eta$), and weight decay ($\lambda$). The results indicate that the best performance is not necessarily obtained at the default momentum values for SGD ($\beta = 0.9$) or Adam ($\beta_1 = 0.9, \beta_2 = 0.99$). For Muon, a higher $\beta$ is generally better for each learning rate, although performance tends to fall for $\beta > 0.9$, except in the case of small learning rates like $\eta = 10^{-5}, 10^{-4}$ where performance remains high. The performance of Shampoo is highly stable across different $\beta_1$ and $\beta_2$ values for any learning rate $\eta$, in sharp contrast to Adam.

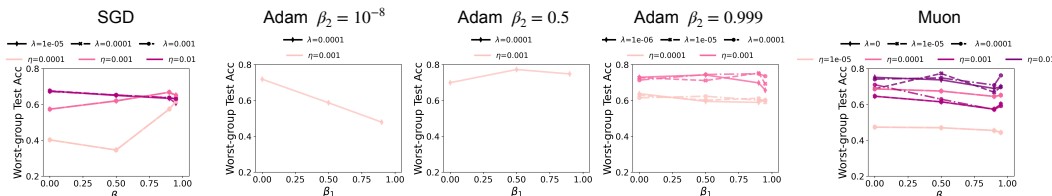

Figure 24: Hyperparameter sweep for different optimizers on the CelebA dataset, showing worst-group test accuracy at the last fine-tuning epoch. The sweep varies the momentum parameters ($\beta$ for SGD/Muon, $\beta_1, \beta_2$ for Adam/Shampoo), learning rate ($\eta$), and weight decay ($\lambda$).

### D.5  TINYSTORIES

*Dataset and Preprocessing.* We use a subset of the TinyStories corpus (Eldan & Li, 2023), randomly sampling 500 stories from the full dataset as training set. The selected stories are tokenized using SentencePiece (Kudo & Richardson, 2018) with unigram tokenization and vocabulary size 2000. We sampled another 50 stories as validation set.

We define token frequency buckets based on corpus statistics computed over the training set. Tokens are categorized as: (1) "rare" tokens: frequency $\leq$ 50th percentile, (2) "frequent" tokens: frequency $\geq$ 80th percentile. See Fig. 26 (left) for a demonstration of token frequency distribution.

*Evaluation Metrics.* We evaluate models on the validation set every epoch using cross-entropy loss on both training and validation sets, and top-1/top-5 accuracy on the validation set.

*Model Architecture and Training.* We train 4-layer Transformer models with the following specifications: maximum sequence length 64, 4 heads, embedding dimension 256, feed-forward dimension 512, and dropout rate 0.1. The token embeddings and unembeddings are tied. All weights are initialized randomly.

Training is conducted with batch size 32 for 100 epochs using three optimizers: SGD ($\eta = 0.1$, $\beta = 0.9$, weight decay $10^{-4}$), Adam ($\eta = 0.001$, $\beta_1 = 0.9$, $\beta_2 = 0.999$, weight decay $10^{-4}$), and Muon ($\eta = 0.1$, $\beta = 0.95$, no weight decay). Learning rates and momentum values are selected via grid search based on the lowest validation loss (see Fig. 25).

*Results.* As shown in Fig. 26 (right), Muon and Adam converge faster than SGD, with Muon showing rapid loss reduction around epoch 10. Critically, SGD exhibits a large gap between majority and minority token performance, while Muon significantly reduces this gap, achieving more balanced learning across token frequencies. Also see Fig. 9 which compares the validation set accuracy, where the effect is more pronounced.

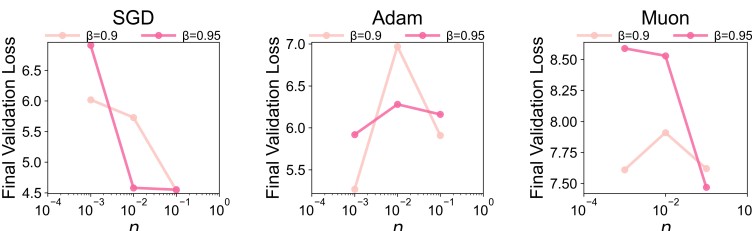

Figure 25: Effect of using different learning rates and momentum on the final validation loss of different optimizers on TinyStories.

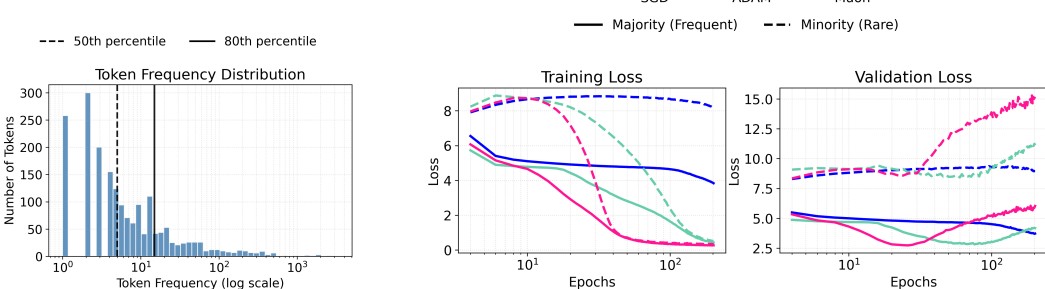

Figure 26: Left: Token frequency distribution in training set. The x-axis shows token rank ordered by frequency (most to least frequent), and the y-axis shows log frequency. Tokens at or below the 50th percentile (left of dashed line) are classified as rare tokens, while tokens at or above the 80th percentile (right of solid line) are classified as frequent tokens. Right: Comparison of train and validation loss dynamics of various optimizers on TinyStories. Solid lines: frequent tokens ($\geq$ 80th percentile), dashed lines: rare tokens ($\leq$ 50th percentile). Muon leads to reduced performance gap between frequent and rare tokens on the train set compared to SGD and Adam.

### D.6 ATTRIBUTE-ORGANISM CLASSIFICATION

We consider structured attribute prediction using an organism dataset to study hierarchical concept learning and the effect of using different optimizers.

*Dataset and Task.* The dataset contains 12 organisms across hierarchical categories: plants (apple, orange, rose, tulip), animals (dog, wolf, cat, tiger, sparrow, eagle, salmon, trout), mammals (dog, wolf, cat, tiger), and other animals. Each organism has 13 attributes with taxonomic structure: plant attributes ("uses sunlight for food," "grows roots"), general animal attributes ("needs oxygen," "can move around"), mammal attributes ("stays warm-blooded"), etc. The conditional probability matrix, given attribute and predict subject, is shown in Fig. 27. This creates a hierarchical classification challenge where the model must learn both individual attribute predictions and the underlying taxonomic relationships.

*Model Architecture and Training.* We use a bilinear model with logits $= W^1 W^0$, where $W^0 \in \mathbb{R}^{d \times n}$ and $W^1 \in \mathbb{R}^{k \times d}$, with $n = 12$ (number of organisms), $k = 13$ (number of attributes), and $d = 128$ hidden dimensions. GD uses learning rate $\eta = 0.001$ and Muon uses learning rate 0.01, with Muon using momentum $\beta = 0.9$.

*Results.* Fig. 27 shows confusion matrices tracking category classification for GD and Muon. The categories are organized from coarse to fine in these matrices. Each entry indicates the number of examples in the row category being classified in the column category. The results reveal a key difference in learning dynamics: GD learns the coarse categories (Animals and Plants) much faster than fine-grained categories, showing strong diagonal entries for these broad classes early in training. In contrast, Muon learns all hierarchical levels at similar rates, achieving more balanced progress across both coarse and fine categories. This differential learning behavior directly supports our theoretical finding that spectrum-aware optimizers like Muon learn principal components at equal rates, while GD prioritizes learning dominant patterns first.

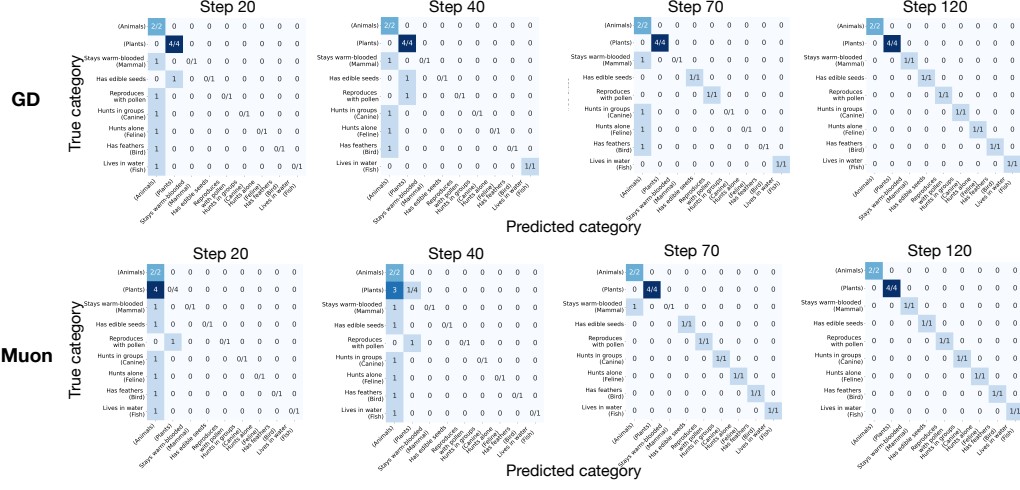

Figure 27: Confusion matrix evolution on attribute classification for GD and Muon. Categories are organized from coarse (top-left) to fine (bottom-right). Each cell value indicates the number of attributes with true category (row) classified as predicted category (column). A diagonal matrix indicates perfect classification. GD learns coarse categories first (evident at step 20), while Muon learns all hierarchical levels at similar rates (step 70). Both optimizers achieve perfect classification by step 120.

### D.7 MIXTURE OF TINYSTORIES AND ADD-$k$

In this section, we investigate whether our findings on imbalanced learning extend to the level of knowledge acquisition and skill learning. We introduce a domain-imbalanced setting by constructing a

composite dataset where the majority component consists of natural language narratives (TinyStories), while the minority component requires learning a structured mathematical skill.

*Dataset and Task.* We train on a mixture of the TinyStories dataset (majority) and a mathematical task (minority), specifically add-$k$. For TinyStories, we follow the same train/test process as in App. D.5. For add-$k$, an example sequence is of the form "Q: $x_1$, A: $y_1$, Q: $x_2$, A: $y_2$, Q: $x_3$, A: $y_3$", where $y_j = x_j + k$. We consider two-digit natural numbers for both the inputs $x_j$ and the offsets $k$, where $k$ is drawn from a fixed set of values during both training and evaluation. We train using the next token prediction objective and evaluate the exact match accuracy to predict the last $y_3$, given the rest of the sequence.

*Model Architecture and Training.* We use the same Transformer architecture as in our TinyStories experiments (see App. D.5 for details). We train on a composite dataset of $20,000$ samples. This includes $18,000$ TinyStories samples (majority, with $0.9$ fraction) and $2,000$ add-$k$ arithmetic sequences (minority, with $0.1$ fraction). We evaluate on $500$ held-out samples for each domain. We train for 200 epochs with a batch size of 256 and compare three optimizers. SGD uses $\eta = 5 \times 10^{-2}$, momentum 0.95, and weight decay $10^{-4}$. For Adam, we set $\eta = 10^{-3}$, $\beta_1 = 0.9, \beta_2 = 0.999$, and weight decay $10^{-4}$. For Muon, we set $\eta = 10^{-2}$ and momentum 0.95.

*Results.* We compare the top-1 and top-5 accuracies on TinyStories and the exact match accuracies on add-$k$ for SGD, Adam, and Muon in Fig. 28. We observe that for Muon, the accuracies on both datasets increase at a more similar rate compared to SGD, and it generalizes better on the minority add-$k$ task. Adam, on the other hand, achieves performance comparable to Muon, similar to some of our results on subgroup robustness datasets in Sec. 4.

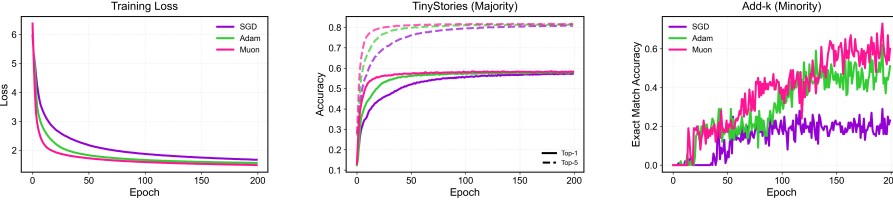

Figure 28: Comparison of SGD, Adam and Muon when training a transformer on a mixture of TinyStories (majority) and add-$k$ (minority) datasets: train loss (left), top-1 and top-5 test accuracies on TinyStories (middle) and exact match test accuracy on add-$k$ (right). Muon promotes more balanced learning compared to SGD while performing comparably to Adam.

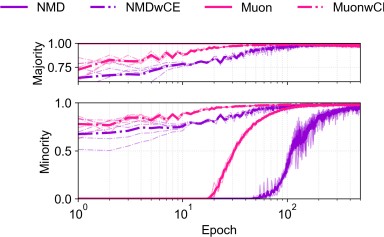
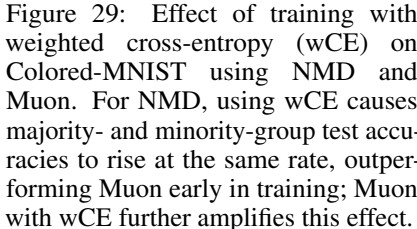

Figure 29: Effect of training with weighted cross-entropy (wCE) on Colored-MNIST using NMD and Muon. For NMD, using wCE causes majority- and minority-group test accuracies to rise at the same rate, outperforming Muon early in training; Muon with wCE further amplifies this effect.

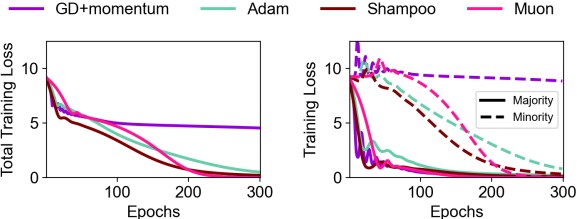

Figure 30: Comparison of GD, Adam, Shampoo and Muon when training a CNN on the Barcoded MNIST dataset from Kunstner et al. (2024), a variant of MNIST with heavy-tailed class imbalance. GD only drives the loss on majority classes towards 0 and makes little progress on the minority classes. In contrast, Adam, Shampoo and Muon drive the loss on both majority and minority classes towards 0.

### D.8 EFFECT OF LOSS RE-WEIGHTING

In Fig. 29, we compare training with weighted cross-entropy (wCE) loss on the Colored-MNIST dataset (same setting as Fig. 1) using NMD and Muon. We find that for NMD with wCE, the majority- and minority-group accuracies rise at the same rate, and it outperforms Muon (with CE) very early in training, while using Muon with wCE further amplifies this effect. Since Muon with CE already exhibits a weaker form of this effect 'for free', it may still be useful in settings where explicit group information for loss re-weighting is unavailable.

### D.9 MNIST WITH HEAVY-TAILED CLASS IMBALANCE

*Dataset.* In this section, we consider the Barcoded MNIST dataset, which is a variant of MNIST with heavy-tailed class imbalance introduced in Kunstner et al. (2024). Barcoded MNIST contains two types of classes: 10 classes with 5000 samples each from the original MNIST dataset (majority), and $10 \times 2^{10}$ additional classes with 5 samples each (minority). The images in the minority classes are generated by taking images from the original MNIST dataset, and encoding a 10-bit barcode into the top left corner of the images, for each of the 10 original classes.

*Model Architecture and Training.* Following Kunstner et al. (2024), we train a 2-layer CNN on this dataset in the full-batch setting. In Fig. 30, we compare the total train loss as well as train loss on the majority and minority classes (each comprising about $\approx 50\%$ of the total samples) for the four optimizers: GD with momentum, Adam, Shampoo and Muon (we use `learning rates` 0.005, $10^{-4}$, $10^{-4}$, and 0.005, respectively). We use default momentum parameter values for all optimizers. For Shampoo, we use `AdamGraftingConfig`, and set the stabilization parameter to $10^{-8}$.

*Results.* Consistent with the results in Kunstner et al. (2024), we observe while that GD only minimizes the loss on the majority classes, and makes negligible progress on the minority classes, Adam minimizes loss on both majority and minority classes. In addition, we find that training with Muon or Shampoo leads to a similar behaviour as Adam: they also minimizes the loss on both majority and minority classes, in contrast to GD.

### D.10 SPECTRAL ANALYSIS FOR NON-LINEAR MODELS

In this section, we extend our analysis on the rate of learning of different spectral components beyond linear models. Specifically, we consider two settings with step imbalance ($k/2$ majority and $k/2$ minority classes) trained using cross-entropy (CE) loss: i) one-hidden-layer MLP (with fixed outer layer, as used in Fig. 1) trained on the data model (DM) with an imbalance ratio of 10, and ii) a ResNet-18 trained on CIFAR-10 with an imbalance ratio of 20 (same setting as App. D.2 for Fig. 3). In both settings, we track the singular values of the $k \times n$ logit matrix using all $n$ train samples across the $k$ classes as training progresses. Specifically, we quantify the extent of balanced learning by measuring the KL divergence between the uniform probability vector $\frac{1}{k}\mathbf{1}$ and the vector of normalized singular values $\frac{\text{diag}(\Sigma_t)}{\text{tr}(\Sigma_t)}$ over training time, where $\Sigma_t$ is the singular value matrix at time $t$.

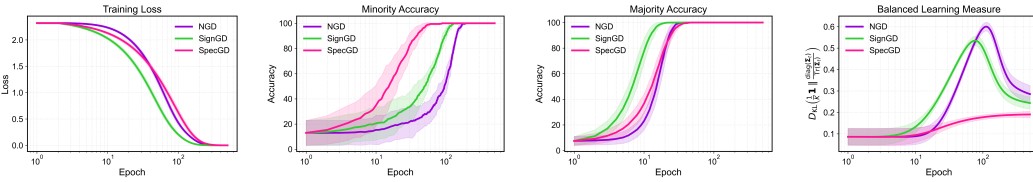

Figure 31: Comparison of NGD, SignGD, and SpecGD on a one-hidden ReLU MLP on Gaussian mixture data generated using the data model Eq. (DM). From left to right: train loss, minority- and majority-class test accuracies, and KL divergence between the normalized all-ones vector and the vector with normalized singular values of the logit matrix over time. SpecGD maintains a consistently low KL divergence throughout training. In contrast, NGD and SignGD exhibit an increase in this metric, indicating a period of spectral imbalance.

**Gaussian Mixture Setting.** We consider $n = 1000, d = 200$. Similar to Fig. 4, we compare optimizers NGD, SignGD and SpecGD in this case, with learning rates $8 \times 10^{-2}, 8 \times 10^{-4}$, and

$2 \times 10^{-2}$, respectively. We use the full batch for training, and use a held out set of 2000 test samples for reporting test accuracies. The results are reported after averaging over 5 independent seeds for initialization and data selection. As shown in Fig. 31, for NGD and SignGD, the KL divergence-based metric initially increases as training progresses, indicating more imbalanced singular values, and becomes smaller in later stages of training, as both majority and minority classes are learned. In contrast, SpecGD maintains a consistently lower value for this metric throughout training, demonstrating that its normalized singular values remain closer to uniform and that spectral components are learned at a more balanced rate. These findings are supported by the corresponding majority- and minority-class accuracies.

**CIFAR-10 with Class Imbalance.** We repeat the experiment described above on the imbalanced CIFAR-10 dataset (same setting as App. D.2 for Fig. 3). In Fig. 32, we observe that Muon similarly demonstrates a more balanced learning of the spectral components. This is supported by the dynamics of majority- and minority-class accuracies (see Fig. 3).

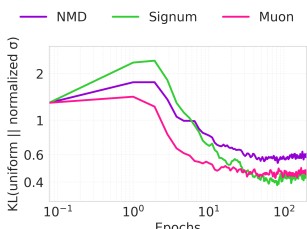

# E RELATED WORK

**Spectrum-aware and Other Adaptive Optimizers.** Our work is situated within the growing body of research on spectrum-aware and matrix-level optimizers, which have demonstrated significant empirical success. The development of methods like Shampoo (Gupta et al., 2018) and, more recently, Muon (Pethick et al., 2025; Jordan et al., 2024) has high-

Figure 32: Comparison of NMD, Signum, and Muon on imbalanced CIFAR-10 showing the KL divergence between the uniform probability vector and the vector of normalized singular values of the logit matrix. Muon maintains a lower KL divergence throughout training compared to NMD and Signum indicating more balanced learning of the spectral components.

lighted the potential of optimizers that operate with matrices at the layer level (Carlson et al., 2015; Large et al., 2024), rather than on vectorized parameters. Recent theoretical inquiries have sought to understand the mechanisms behind their performance. One line of work has focused on characterizing the implicit bias of their canonical form, SpecGD. For instance, Fan et al. (2025); Tsilivis et al. (2024) have shown that SpecGD converges to a max-margin classifier with respect to the spectral norm, though this analysis primarily describes the terminal phase of training. Concurrently, other research has investigated the advantages of adaptive methods like Adam, particularly in settings with heavy-tailed class imbalance (Kunstner et al., 2024) or from the perspective of feature learning (Vasudeva et al., 2025b). While these works highlight Adam's advantages on different forms of imbalanced data through various mechanisms, we use this testbed to reveal a fundamentally different principle behind SpecGD's success: its balanced learning of the data's principal components, which correspond directly to majority and minority groups.

Further, given that our experiments benchmark spectral methods against strong adaptive baselines, it is worth noting that recent large-scale empirical studies on language model pretraining confirm that well-tuned Adam variants remain highly competitive. Extensive benchmarks have shown that AdamW is a robust and strong baseline (Wen et al., 2025; Semenov et al., 2025), which is consistent with our experiments in Sec. 4, where Adam is on par with or sometimes beats Shampoo and Muon. Orvieto & Gower (2025) showed, via extensive hyperparameter tuning, that $\beta_1 = \beta_2$ is a near-optimal choice for momentum parameters in Adam, contrary to conventional wisdom. They also showed that Signum recovers much of the performance gap between SignGD and Adam, a trend we also observe in our experiments (Fig. 15). A key factor in its performance relative to SGD is the batch size, as the Adam-SGD gap is most pronounced in large-batch training regimes (Srećković et al., 2025; Marek et al., 2025). In line with these works, we tune hyperparameters such as the learning rate and momentum parameters for all our experiments (see App. D).

**Training Dynamics.** Our theoretical analysis of training dynamics builds directly upon the framework established by Saxe et al. (2013) and subsequently extended by Gidel et al. (2019). These works provided exact, closed-form solutions for the learning dynamics of deep linear networks trained with standard GD. Their key insight was to analyze the system under the condition that the input and output moment matrices are jointly diagonalizable, which simplifies the complex, non-linear dynamics into a set of decoupled scalar equations. While their analysis revealed how GD prioritizes

learning the dominant spectral components of the data, our work adopts this lens and extends it to SpecGD. This extension allows for a direct, analytical comparison between the training trajectories of GD and SpecGD, enabling us to precisely characterize the latter's more balanced learning rate across all spectral components. Furthermore, our analysis of the bilinear model (Sec. 3.3) also connects to the literature on the Unconstrained Features Model (UFM), which is often used as a theoretical proxy to study the geometry of feature learning in deep networks, positing a model with enough capacity to freely optimize both the learned representations and the final classification layer (Yang et al., 2018; Mixon et al., 2020).

**Spectral Bias in Deep Learning.** More broadly, our work connects to the well-documented phenomenon of *spectral bias* or *simplicity bias* in deep learning, where networks trained with (S)GD show a predisposition to learn simpler patterns first. Prior works have demonstrated that networks prioritize learning low-frequency functions before high-frequency ones (Rahaman et al., 2019; John Xu et al., 2020), and that SGD guides networks to learn functions of progressively increasing complexity (Nakkiran et al., 2019; Cao et al., 2020). This principle extends even to modern architectures, with recent work showing that Transformers also exhibit a bias towards learning low-sensitivity functions (Bhattamishra et al., 2023; Vasudeva et al., 2025a). However, this inherent bias towards simplicity can be detrimental, as it can lead models to rely on simple but spurious correlations in the data, which hinders generalization on underrepresented groups (Shah et al., 2020; Vasudeva et al., 2025b). Our analysis complements this line of work by showing how spectrum-aware optimizers offer a mechanism to somewhat mitigate this bias by promoting more uniform learning across the spectral components of the data.

## F    LIMITATIONS AND OPEN QUESTIONS

Our theoretical results are derived in a population setting with squared loss, and extending them to the more practical finite-sample regime is a key direction. In this setting, where many interpolating solutions may exist, can we characterize the implicit bias of SpecGD and establish formal generalization guarantees analogous to those in our population-level analysis? Another crucial extension is to move beyond squared loss to analyze commonly used losses like cross-entropy. A primary challenge here is that our key technical assumption of joint diagonalizability may no longer hold, requiring new analytical tools. Furthermore, while our experiments on language modeling (Sec. 4) suggest that Muon's balanced learning persists (higher top-$k$ validation accuracy on rare tokens), a formal extension of our theory is non-trivial. It remains an open question what data structures, analogous to the moment matrices $\Sigma_{xx}$ and $\Sigma_{yx}$ in our multi-class classification setup, define the principal components when doing next-token prediction with sequential, language data. Answering these questions would provide a more complete picture of why spectrum-aware optimizers succeed in modern deep learning.

