# OpenReview forum: "How Muon’s Spectral Design Benefits Generalization: A Study on Imbalanced Data"
_ICLR.cc/2026/Conference — ICLR 2026 Poster_

### Official Review · Reviewer_oR6c · 2025-10-19

**Soundness:** 3
**Presentation:** 3
**Contribution:** 3
**Rating:** 4
**Confidence:** 3

**Summary:**

This paper studies a simplified algorithm spectral GD, as a canonical algorithm to understand algorithms such as Muon, Shampoo, and SOAP. They adopt a linear model under squared-loss, using imbalanced classes as their data, where these classes have orthogonal means, and employ continuous dynamics or discrete dynamics that mimic the continuous case. The results show that for worst-class and class-balanced risks, when considering early phase loss decay, specGD is faster than GD with or without adaptive step-size (normalized gradient). They also prove that depth can accelerate the learning of minor components. Experiments are also provided under practical settings to validate the theoretical findings.

**Strengths:**

The paper provides a relatively thorough analysis of Spectral GD under imbalanced datasets, characterizing both the loss decay and the impact of network depth. The problem they studied is important towards understanding real applications, although under simplied settings. There is both theoretical and empirical evidence to support the claim.

**Weaknesses:**

The overall concern is that the setting of this paper is too strong, More precisely, the authors adopt the specGD, which is a perfectly preconditioned algorithm, while in practice, the main difficulty of adaptive algorithms is to design the preconditioning, and different preconditionings lead to different behavior.

Regarding the theory, the assumptions seem pretty strong. The authors use linear model, orthogonal data, joint diagonalizability (another strong “orghogonality” assumption on data), zero (or near-zero) initialization, and gradient flow (or near-gradient flow).

**Questions:**

1. Please clarify the theoretical relationship between Spectral GD and practical adaptive methods such as Muon, Shampoo, and SOAP. In particular, does Spectral GD explicitly approximate their behavior along the dominant eigen-directions of the preconditioning matrix?

2. Equation (1): The definition of the inner product is unclear. Since \nabla L should have the same dimensions as W_t, the resulting inner product seems dimensionally inconsistent. Please provide a precise definition.

3. It would be helpful to explicitly present the update rules (for NGD, specGD, GD, and their gradient-flow versions) used in Theorems 1 and 2 for clarity.

4. The early phase and asymptotic behavior of specGD and (N)GD are different. It would be better if both can be included in the theorem to give more complete picture of these algorithms.

5. Section 3.3: Proposition 2 and its counterpart for the deep linear model do not seem clear.

5.1 The requirements on \eta are not clearly stated. Theorems 1 and 2 analyze gradient flow, whereas Proposition 2 uses gradient descent. Please clarify the rationale and distinction between these settings.

5.2 The results hold when \delta\to\infty, which makes the current statement a bit wierd since you can then just assume zero initialization. The statement should be rewritten.

5.3 The counterpart result for deep linear models is not formulated as a proposition. It may be clearer to combine these results into a single theorem and then discuss the role of depth.

5.4 The variable t appears in the results, but as t \to \infty, \bar{W} diverges.

5.5 There is a \Delta T that directly characterizes the time of learning different classes but this is not included in any theorems or propositions.

5.6 Overall, section 3.3 needs to be reorganized.

6. Experiments compare toy algorithms (canonical algorithms) for theory, and the practical algorithms for real applications. However, it is unclear whether specGD can adequately represent Muon and Shampoo in practice, given their distinct preconditioning mechanisms. It would be better to include additional experiments that directly compare specGD with Muon and Shampoo for validation.

---

> ### Author Response · Authors · 2025-11-21
>
> Thank you for your time and helpful suggestions to improve our work. We are encouraged that you find our analysis of spectral GD under imbalanced settings thorough, the studied problem important for understanding applications in real settings, and our claims supported by both theory and experiments.
>
> We respond to specific comments below and have **updated the paper** to incorporate your suggestions.
>
> > Q2: Precise definition of inner product.
>
> For matrices $A$, $B$, the inner product $\langle A,B \rangle$ is trace($AB^\top$). We have added this clarification in Section 3 in the paper.
>
> > W1&Q1: “Please clarify the theoretical relationship between Spectral GD and practical adaptive methods such as Muon and Shampoo.”
>
> Thanks for the comment. We have updated **Section 3** of the paper to clarify the connection between SpecGD and Muon, and include details for Shampoo in **App. A**. We briefly explain the connections below.
>
> First, note that spectral GD is steepest descent with respect to spectral norm. Formally, if the gradient at iteration $t$ is $G_t$, GD update uses $\Delta_t=G_t$. Normalized steepest descent update with spectral norm uses $\Delta_t=\arg\max_{\||\Delta\||_2\leq 1} \langle \Delta, G_t \rangle$, where $\||\cdot\||_2$ denotes the spectral norm. Let the SVD of $G_t$ be $U_t\Sigma_tV_t^\top$, then the update $\Delta_t=U_tV_t^\top$.
>
> Next, consider Muon. Conceptually, Muon is steepest momentum descent with respect to spectral norm. Formally, consider momentum $M_t=\beta M_{t-1}+(1-\beta)G_t$ using parameter $\beta$. Then, if the SVD of $M_t$ is $U_t\Sigma_tV_t^\top$, then the update $\Delta_t=U_tV_t^\top$. In practice, Muon does this matrix whitening step by running some iterations of Newton-Schulz on $M_t$. When we consider perfect matrix operations, and set $\beta=0$, Muon reduces to SpecGD.
>
> Finally, consider Shampoo. Define the preconditioning matrices
> $L_t=\beta_2 L_{t-1}+(1-\beta_2)G_tG_t^\top$ and $R_{t}=\beta_2 R_{t-1}+(1-\beta_2)G_t^\top G_t$,
> where the parameter $\beta_2$ denotes the preconditioning accumulation parameter.
>
> Using these preconditioners and the momentum term $M_t = \beta_1 M_{t-1}+(1-\beta_1)G_t$, we get the update $L_t^{-1/4}M_tR_t^{-1/4}$. When $\beta_1=\beta_2=0$, Shampoo reduces to SpecGD (please see App. A in the paper for details).
>
> > W1&Q1: “Does Spectral GD explicitly approximate the behavior of Muon along the dominant eigen-directions of the preconditioning matrix?”
>
> Good question! In order to validate if Muon exhibits learning dynamics similar to SpecGD, which learns different spectral components of the data at the same rate, we have **added two experiments** in the paper.
>
> First, we consider the setting of Fig. 2, where we show the dynamics of a linear model optimized on MSE on the population for our Gaussian mixture data model with class imbalance from zero initialization, and how SpecGD learns classes at the same rate whereas (N)GD priorities learning majorities first. In this setting, we run Muon with default momentum parameter 0.9 and show, in **Fig. 10 in App. C**, that the dynamics closely match those of SpecGD.
>
> Second, we also compare the dynamics of SpecGD and Muon in this setting with small random initialization instead of zero initialization and using finite samples instead of the population setting. In this setting, the Condition 1, under which we prove our theoretical results, does not hold exactly, and we compare the dynamics of Muon to the dynamics of SpecGD both as seen empirically and those predicted by the theory. The results are shown in **Fig. 12** and additional details are included in **App. C** in the paper. We find that in this setting as well, Muon promotes balanced learning of the spectral components and closely matches the SpecGD dynamics.
>
> > Q6: “It is unclear whether specGD can adequately represent Muon and Shampoo in practice, given their distinct preconditioning mechanisms. It would be better to include additional experiments that directly compare specGD with Muon and Shampoo for validation.”
>
> Thank you for the question. In our experiments on the two subgroup robustness benchmark datasets in Section 4, we compare the effect of the momentum parameters for Muon and Shampoo on the generalization performance. Notably, this already includes SpecGD (Muon and Shampoo with the $\beta$ parameters close to 0), and these results are included in **App. D.4** in the paper. Overall, for these datasets, from **Figs. 23 and 24**, we observe that for Shampoo, the momentum parameters do not significantly impact performance for most learning rate values. For Muon, the performance for larger momentum values is comparable or slightly better than that of SpecGD.

---

> ### Author Response · Authors · 2025-11-21
>
> > W2: “Regarding theory, assumptions seem pretty strong. The authors use linear model, orthogonal data, joint diagonalizability (another strong "orthogonality" data assumption), zero (or near-zero) initialization, and gradient flow (or near-gradient flow).”
>
> Thanks for the comment. We note that several of these assumptions are quite standard [1-2] for analyzing GD dynamics, and hence this serves as a reasonable setting for a first step towards theoretically understanding and contrasting SpecGD dynamics with GD. We agree that it is important to extend our analysis in more general settings, and we believe this is a promising direction for future work.
>
> Motivated by your comment, we have **included some experimental results** to validate that our results still hold when some of the conditions are relaxed, as discussed below.
>
> **i) Data model and Joint Diagonalizability condition**: We consider the data model (DM) mainly to illustrate a setting where Condition 1 (joint diagonalizability of the two moment matrices $\mathbb{E}[xx^\top]$ and $\mathbb{E}[yx^\top]$) strictly holds. Our theoretical results hold under other data models as long as Condition 1 is satisfied.
>
> Regarding how reasonable Condition 1 is, note that in [1], the authors validate whether a weaker version of Condition 1 is satisfied for some datasets used in practice. Specifically, they check whether there exist orthonormal matrices $U, V$ such that the empirical moment matrices are jointly diagonalizable as $V(S_{xx}+B)V^\top$ and $US_{yx}V^\top$, respectively, where $\||B\||$ is small with respect to $\||S_{xx}\||$. They find that on datasets like MNIST, CIFAR, etc., this ratio is small (see $\Delta_{xy}$ in Table 1 in [1]), so this can be considered as a reasonable condition.
>
> Further, for our data model, it can be shown that when the requirement of orthogonal means is relaxed (e.g., mean vectors are sampled from a random Gaussian, so under high dimensions, $|\bar{\mu}_i^\top\bar{\mu}_j|$ for $i\neq j$ is small with high probability), the population moment matrices in this case also satisfy the weaker condition.
>
> We consider Condition 1, where $\||B\||=0$, mainly to make the analysis more tractable. In **App. C (Fig. 13)**, we have added an experiment where we empirically show that GD/SpecGD dynamics predicted by the theory closely track the dynamics in a setting where only the weaker version holds.
>
> **ii) Initialization**: We have added an experiment in **App. C (Fig. 11)**, where we show that GD/SpecGD dynamics predicted by the theory (zero initialization) closely track the dynamics in a setting with small random Gaussian initialization.
>
> **iii) Gradient flow**: Note that our results characterizing SpecGD dynamics for both linear and deep linear models (Props. 1-4) hold for both discrete and continuous time. While our results in Theorems 1 and 2 contrasting SpecGD vs GD and NGD are stated using the flow versions, our experiments show the same behaviours using discrete time updates.
>
> References:
>
> [1] Gidel, G., Bach, F., and Lacoste-Julien, S. Implicit regularization of discrete gradient dynamics in linear neural networks. NeurIPS 2019.
>
> [2] Saxe, A. M., McClelland, J. L., and Ganguli, S. Exact solutions to the nonlinear dynamics of learning in deep linear neural networks. 2013.
>
> > Q5: Comments regarding Section 3.3
>
> Thank you for your feedback to help improve this section.
>
> > 5.1 “Theorems 1 and 2 analyze gradient flow, whereas Proposition 2 uses gradient descent. Please clarify the rationale and distinction between these settings.”
>
> Theorems 1 and 2 compare different optimizers (SpecGF vs GF and NGF), hence we use the flow versions which are step-size independent, for fair comparison. While stating the dynamics of SpecGD, we consider the discrete-time versions (Prop. 1 for linear models and Prop 2 for two-layer models).
>
> Note that, in App. B.2, we include the proofs for both SpecGD and SpecGF dynamics for linear models. We can similarly obtain the continuous-time SpecGF dynamics for two-layer linear models.
>
> > 5.2 “The results hold when \delta\to\infty, which makes the current statement a bit weird since you can then just assume zero initialization. The statement should be rewritten.”
>
> We note that the dynamics of SpecGD for deep linear models can be exactly characterized for any $\delta>0$, as mentioned in both Prop. 3 in App. B.5 (full formal version of Prop. 2 for two-layer models) and Prop. 4 in App. B.6 (deeper models). In the main body, we state a simplified version of these results with $\delta\to\infty$, to ignore the initialization terms, and give cleaner expressions for the dynamics. We have added a small clarification for this after Prop. 2.
>
> Additionally, note that for models with two or more layers, the gradients for each layer depend on other layers. This can be seen from the gradient expressions in the proofs in App. B.5 and B.6. Therefore, we can only use a small-scale initialization, not exact zero initialization.

---

> ### Author Response · Authors · 2025-11-21
>
> > 5.3 “The counterpart result for deep linear models is not formulated as a proposition. It may be clearer to combine these results into a single theorem and then discuss the role of depth.”
>
> As mentioned in the paper, the full formal statement of the result for deep linear models is included as Prop. 4 in App. B.6. We did not include the full statement in Section 3.3 due to space constraints.
>
> We present the result for depth $L=2$ linear model separately first for two reasons. First, prior work [1-2] characterized GD dynamics for two-layer models, and we wanted to contrast our results characterizing specGD dynamics with those of GD. Second, as mentioned in Section 3.3 in the paper, two-layer linear models are closely connected to the unconstrained features model (UFM), which has been adopted extensively to theoretically study the geometry of representations in deep neural networks [3-4].
>
> References:
>
> [1] Gidel, G., Bach, F., and Lacoste-Julien, S. Implicit regularization of discrete gradient dynamics in linear neural networks. NeurIPS 2019.
>
> [2] Saxe, A. M., McClelland, J. L., and Ganguli, S. Exact solutions to the nonlinear dynamics of learning in deep linear neural networks. 2013.
>
> [3] Yang, Z., Dai, Z., Salakhutdinov, R., and Cohen, W. W. Breaking the softmax bottleneck: A high-rank rnn language model. 2018.
>
> [4] Mixon, D. G., Parshall, H., and Pi, J. Neural collapse with unconstrained features. 2020.
>
> > 5.4 “The variable t appears in the results, but as t \to \infty, \bar{W} diverges.”
>
> Note that in our theoretical setting, there is a unique solution $W^*=US_{xx}^{-1}S_{yx}V^\top$ and all optimizers converge to this solution in finite time and iterate norms always remain bounded.
>
> > 5.5 “There is a \Delta T that directly characterizes the time of learning different classes but this is not included in any theorems or propositions.”
>
> As suggested, we have added $\Delta T$ in Prop. 4 in the Appendix, which is the full formal statement of our result characterizing the dynamics of SpecGD for deep linear models.
>
> > Q4: “The early phase and asymptotic behavior of specGD and (N)GD are different. It would be better if both can be included in the theorem to give a more complete picture of these algorithms.”
>
> Thanks for the suggestion. We have added a comment stating that different optimizers eventually converge to the same solution, in the paragraph before Theorem 2.
>
> > Q3: “It would be helpful to explicitly present the update rules (for NGD, specGD, GD, and their gradient-flow versions) used in Theorems 1 and 2 for clarity.”
>
> Thanks for the suggestion, we have added the updates for NGD and SpecGD in **Section 3** in the paper, and the gradient-flow versions in App. B.2 due to space constraints.
>
> We hope that these responses help address your concerns and that you would consider increasing your score.

---

> > ### Comment · Reviewer_oR6c · 2025-11-26
> >
> > I would like to thank the authors for the reply and revision. The relationship between specGD and practical algorithms has now been empirically clarified. All my other questions have been addressed. Overall, the theoretical settings of this paper are still strong, but there are empirical experiments that complement and validate the main claim, and the theoretical insights are good. I will increase my score.

---

> > > ### Author Response · Authors · 2025-11-27
> > >
> > > Thank you again for your feedback to help improve the paper, and for increasing the score. We appreciate your positive endorsement and support.

---

### Official Review · Reviewer_v3RF · 2025-10-28

**Soundness:** 3
**Presentation:** 3
**Contribution:** 2
**Rating:** 2
**Confidence:** 4

**Summary:**

This paper verified that spectral gradient descent learns all principal components of the data at equal rates, especially for imbalanced datasets. The conclusion extends to deep linear models and spectral optimizers including Muon and Shampoo, where the study concluded that these optimizers achieve better generalization through the balanced learning rates.

**Strengths:**

1. The study promotes the understanding of the success of spectral optimizers from a fundamental perspective, especially when the datasets are imbalanced in class or group.
2. The studies in this paper convered a wide range of neural architectures. The experimental designs are trustworthy.
3. Good story-telling from multiple perspectives.

**Weaknesses:**

1. This paper frames SpecGD as the canonical form for Muon and Shampoo. However, the practical implementations involve crucial components like preconditioner history accumulation  and various approximations/stabilization techniques. The paper could benefit from a more detailed discussion of how these practical elements might alter the "equal-rate" learning dynamics of the idealized SpecGD.
2. The extended analytics to language modeling (next token prediction) is limited to rare and frequent tokens learning, while data imbalance on linguistics or knowledge and skills are investigated less in the study.
3. Adam and other second-order preconditioning approximators should also be studied in simple linear models and image classification tasks, as its relation to second-order curvatures and simplicity in implementation and computation are still critical perspectives at present.

**Questions:**

1. The argument in Section 3.4 relies on assumptions of similar scaling between Hessian blocks (Theorem 6).  Could you comment on how violations of this assumption might affect the results? For instance, in real networks, do earlier layers and later layers exhibit vastly different spectral norms, and how would that impact the perturbation analysis?
2. Is it possible to construct an artifical dataset with a small proportion of domain-specific text corpus, e.g. mathematics or coding, and investigate if Muon exhibits a stronger learning progression in the domain, and if the learning dynamics generalizes to a knowledge-level?
3. While spectral optimizers may promote equal learning on imbalanced datasets, does the conclusion leads to a hypothesis that an ideal data preprocessing algorithm is equivalent to adopting Muon to Adam?

---

> ### Author Response · Authors · 2025-11-21
>
> Thank you for your time and helpful suggestions to improve our work. We are encouraged that you appreciate the paper’s narrative and experimental design and contribution towards better understanding spectrum-aware optimizers from a fundamental perspective.
>
> We respond to specific comments below and have **updated the paper** to incorporate your suggestions.
>
> > W1: “The paper could benefit from a more detailed discussion of how the practical elements like preconditioner history accumulation and various approximations/stabilization techniques might alter the "equal-rate" learning dynamics of the idealized SpecGD.”
>
> That’s a good point! We have **added two experiments** to address this in the paper.
>
> First, we consider the setting of Fig. 2, where we show the dynamics of a linear model optimized on MSE on the population for our Gaussian mixture data model with class imbalance from zero initialization, and how SpecGD learns classes at the same rate whereas (N)GD priorities learning majorities first. In this setting, we run Muon with default momentum parameter 0.9 and show, in **Fig. 10 in App. C**, that the dynamics closely match those of SpecGD.
>
> Second, we also compare the dynamics of SpecGD and Muon in this setting with small random initialization instead of zero initialization and using finite samples instead of the population setting. In this setting, the Condition 1, under which we prove our theoretical results, does not hold exactly, and we compare the dynamics of Muon to the dynamics of SpecGD both as seen empirically and those predicted by the theory. The results are shown in **Fig. 12** and additional details are included in **App. C** in the paper. We find that in this setting as well, Muon promotes balanced learning of the spectral components and closely matches the SpecGD dynamics.
>
> We have added pointers to these experiments in **Section 3.2** in the paper.
>
> Additionally, for our experiments on the two subgroup robustness benchmark datasets in Section 4, we compare the effect of the momentum parameters for Muon and Shampoo on the generalization performance. These results are included in **Figs. 23 and 24 in App. D.4** in the paper.
>
>
> > W3: “Adam and other second-order preconditioning approximators should also be studied in simple linear models and image classification tasks.”
>
> Thank you for raising this point. As suggested, for our *linear multiclass classification* setting in Fig. 2 with Gaussian mixture data, we have added comparisons with Adam (using default $\beta_1,\beta_2$ values 0.9, 0.999, respectively) and SignGD (the canonical form of Adam with $\beta_1=\beta_2=0$), in **Fig. 10 in App. C** in the paper. As noted in the response to W1 above, we have also added Muon for comparison in this figure. Overall, we find that with SignGD and Adam, the classes are learned at a more balanced rate compared to (N)GD, but not at a strictly equal rate as with SpecGD.
>
> Further, we note that in our experiments with *image classification* datasets in Figs. 1 and 3, we compare Muon (steepest momentum descent with spectral norm) to Signum (steepest momentum descent with $\max$ norm). We compare with Signum instead of Adam here, to examine the effect of the geometry of the norm constraint (spectral vs max), in isolation from the effect of using preconditioner accumulations (as used in Adam). For completeness, we do include comparisons with Adam in these settings in **App. D.1 (Fig. 15) and D.2 (Figs. 18 and 19)**, respectively. We have updated **Section 3 and App. A** in the paper to clarify the connections between different optimizers.
>
> > W2, Q2: “Is it possible to construct an artificial dataset with a small proportion of domain-specific text corpus, e.g. mathematics or coding, and investigate if Muon exhibits a stronger learning progression in the domain, and if the learning dynamics generalizes to a knowledge-level?”
>
> Thanks for the suggestion. We have added an experiment in **App. D.7 (Fig. 28)** in the paper, where we train on a mixture of TinyStories dataset (majority) and a mathematical task (minority), namely add-$k$, where an example sequence is “Q: $x_1$, A: $y_1$, Q: $x_2$, A: $y_2$, Q: $x_3$, A: $y_3$”, where $y_j=x_j+k$ (see App. D.7 for further details). We compare the top-1 and top-5 accuracies on TinyStories and exact match accuracies on add-$k$ (to predict the last $y_3$, given the rest of the sequence), for SGD, Adam and Muon. We observe that for Muon, the accuracies on both datasets increase at a more similar rate compared to SGD, and it generalizes better on the minority add-$k$ task. Adam, on the other hand, achieves performance comparable to Muon.

---

> ### Author Response · Authors · 2025-11-21
>
> > Q3: “While spectral optimizers may promote equal learning on imbalanced datasets, does the conclusion lead to a hypothesis that an ideal data preprocessing algorithm is equivalent to adopting Muon to Adam?”
>
> Thank you for the question. Regarding comparison of Muon and Adam, in our experiments (in Sec. 4), we find sometimes Muon may be better than Adam, but in some cases Adam can outperform Muon. Our main claim is that spectrum-aware optimizers, like SpecGD or Muon generalize better than their Euclidean counterparts, like NMD or SGD in imbalanced settings. Now, regarding your comment about the effect of a training intervention, in our paper, we checked the effect of simple loss re-weighting in **App. D.8 (Fig. 29)**. We compared training with NMD using weighted cross entropy loss where samples from minority groups/classes are assigned larger weights vs training with Muon using standard cross entropy loss, on the Colored-MNIST dataset. From the results in **Fig. 29**, we find that loss re-weighting with NMD helps: the minority and majority group test accuracies increase at similar rates. Note that while loss reweighting requires information about group labels, Muon, on the other hand, implicitly promotes balanced learning. Additionally, using loss-reweighting with Muon further amplifies this effect.
>
>
> > Q1: “The argument in Section 3.4 relies on assumptions of similar scaling between Hessian blocks (Theorem 6)....”
>
> We are unsure how to address this question, as our submission does not contain a Section 3.4, a Theorem 6, or any assumptions related to the Hessian. It would be helpful if you could clarify which part of the paper this refers to, so that we can respond appropriately.
>
> We hope that these responses help address your concerns and that you would consider increasing your score.

---

### Official Review · Reviewer_HTPs · 2025-10-31

**Soundness:** 3
**Presentation:** 3
**Contribution:** 3
**Rating:** 6
**Confidence:** 4

**Summary:**

This paper investigates when spectrum-aware optimizers like Muon and Shampoo generalize better than standard gradient descent methods by analyzing their canonical form, Spectral Gradient Descent (SpecGD). Using imbalanced data as a testbed under a mixture-of-gaussians data model, the authors prove for a linear model that SpecGD learns all principal components of the data at equal rates, unlike gradient descent which prioritizes dominant components. Paper shows that this leads to superior balanced accuracy early in training for linear and deep linear models under a Gaussian mixture data model. Authors also present experiments across image classification, natural language inference, and language modeling tasks demonstrate that Muon achieves faster generalization on minority classes and rare tokens compared to SGD, though Adam sometimes matches or exceeds Muon's performance.

**Strengths:**

1. The paper presents a clear empirical observation (Fig. 1) to motivate imbalanced data as a testbed, which is an effective approach.
2. The writing is clear, and the experiments cleanly motivate the theoretical arguments. The paper provides valuable insight in Theorem 1 on the gap between gradient descent and spectral GD (specifically between their continuous flow counterparts), showing that spectral GD outperforms GD.
3. The analysis extends beyond a single-layer linear network to a sequence of matrices, creating a deep linear network framework.
4. Experiments span several tasks (image classification, sentence relationship classification, next-token prediction), datasets (Dominoes, MultiNLI, TinyStories), and architectures, demonstrating that worst-group accuracy is generally learned faster with Muon. This is an interesting observation across various settings.

**Weaknesses:**

1. In the data model (DM), requiring orthogonal $\mu_i$ seems like a strong assumption. It is unclear whether orthogonal means for Gaussian mixtures is a standard assumption. What breaks down if this assumption is removed? In fact, in line 173, experiments are done with $\mu_i$ sampled uniformly from a Gaussian, so only $\mathbb{E}[\mu_i, \mu_j] = 0$ holds.
2. Why is joint diagonalizability a useful condition for the analysis? More importantly, are there other data models, theoretically or empirically, that would satisfy this condition? When does it break down? The paper does not discuss this condition extensively, but it would be briefly useful to discuss it in the text.
3. I am slightly confused about the takeaways for neural network training with non-linear activations. The analysis would be significantly harder, but the paper has a disconnect between presenting closed-form training dynamics for a linear model on a special data model (DM) and the motivation and experiments for deep neural networks on general image classification datasets. SVD analysis works for MSE minimization in a linear model, but would this be useful even in two-layer networks with a nonlinear activation?
4. In Proposition 2, the initialization for matrices is extremely specific, and it is unclear what all the values mean. This is unlike Proposition 1 and Theorem 1, where the initialization was zero. This makes me question the effect of initialization on the results.
5. Like the spectral dynamics visualized and derived for linear networks, the paper would benefit from a similar analysis in the experiment section that goes beyond just worst-group accuracy with different optimizers.

**Questions:**

1. Since the authors are arguing that Muon/Shampoo perform differently than Euclidean methods, can they add Adam to Fig. 1 (left) as well? Also add it to Fig. 3 and others. I am not sure what to read from the eigenvalue and eigenvector figures.
2. Is there any insight into what about the class imbalance/spectra dictates the speedup in generalizing on minority classes with Muon?
3. In Fig. 4, what is the structure of the linear model—just $Wx$?
4. In Proposition 1, there is a dependence on the iterate $t$ and step size $\eta$. What are the implications of the step size value?
5. A stretch question: why is momentum in Adam insufficient to leverage spectral information in the dataset? Experiments do show that Muon is not always better than Adam, e.g., in Fig. 8 at the end.

### Typos and Editorial Suggestions
1. The paper notes that Adam reduces to SignGD without momentum and preconditioning histories, which has been helpful for understanding Adam. Can the authors briefly mention here how Shampoo/Muon reduce to SpecGD and under what assumptions? This would make the writing parallel because in line 73, the authors argue for studying SpecGD as the canonical form for Shampoo and Muon.

---

> ### Author Response · Authors · 2025-11-21
>
> Thank you for your detailed review and overall positive feedback. We are encouraged that you appreciate the effectiveness of using imbalanced data as a testbed, the value of our theoretical results showing how SpecGD can outperform GD, and the clear connection between the experiments and theory. We are glad that you find our observations about Muon’s generalization interesting and recognize the breadth of our experiments across multiple tasks and datasets.
>
> We respond to specific comments below and have **updated the paper** to incorporate your suggestions.
>
> > W1&W2: “In the data model (DM), requiring orthogonal means seems like a strong assumption. What breaks down if this assumption is removed? Are there other data models, theoretically or empirically, that would satisfy the joint diagonalizability condition? When does it break down? Why is it a useful condition for the analysis? It would be useful to discuss this condition in the text.”
>
> Thank you for these great questions. We consider the data model (DM) to illustrate a setting where Condition 1 (joint diagonalizability of the two moment matrices $\mathbb{E}[xx^\top]$ and $\mathbb{E}[yx^\top]$) strictly holds. In [1], the authors validate whether a weaker version of Condition 1 is satisfied for some datasets used in practice. Specifically, they check whether there exist orthonormal matrices $U, V$ such that the empirical moment matrices are jointly diagonalizable as $V(S_{xx}+B)V^\top$ and $US_{yx}V^\top$, respectively, where $\||B\||$ is small with respect to $\||S_{xx}\||$. They find that on datasets like MNIST, CIFAR, etc., this ratio is small (see $\Delta_{xy}$ in Table 1 in [1]), so this can be considered as a reasonable condition.
>
> For our data model, it can be shown that the empirical moment matrices (using finite samples instead of population), also satisfy this (weaker) condition. Similarly, when the requirement of orthogonal means is relaxed (e.g., mean vectors are sampled from a random Gaussian, so under high dimensions, $|\bar{\mu}_i^\top\bar{\mu}_j|$ for $i \neq j$ is small with high probability), it can be shown that the population moment matrices in this case also satisfy the weaker condition.
>
> We consider Condition 1, where $\||B\||=0$, mainly to make the analysis more tractable. In **App. C (Fig. 13)** in the paper, we have added an experiment where we empirically show that GD/SpecGD dynamics predicted by the theory closely track the dynamics in a setting where only the weaker version holds. We have also **updated Section 3.2** to discuss this, as per your suggestion.
>
> References:
>
> [1] Gidel, G., Bach, F., and Lacoste-Julien, S. Implicit regularization of discrete gradient dynamics in linear neural networks. NeurIPS 2019.
>
> > W3: “…the paper has a disconnect between presenting closed-form training dynamics for a linear model on a special data model (DM) and the motivation and experiments for deep neural networks on general image classification datasets. SVD analysis works for MSE minimization in a linear model, but would this be useful even in two-layer networks with a nonlinear activation?”
>
> Thanks for the comment. Motivated by your questions, we have added an experiment in **App. D.10 (Fig. 31)** in the paper. We train a one-hidden-layer MLP on Gaussian mixture data generated from our data model (DM) with cross-entropy loss using different optimizers (NGD, SignGD, SpecGD), and track the singular values of the $k\times n$ logit matrix using all $n$ train samples across $k/2$ majority and $k/2$ minority classes as training progresses. Specifically, to quantify the extent of balanced learning, we track the KL divergence between the uniform probability vector and the vector of normalized singular values over training time. We observe that for NGD and SignGD, the metric first increases as training progresses, indicating more imbalanced singular values, and becomes smaller in later stages of training, as both majority and minority classes are learned. In contrast, SpecGD maintains a consistently lower value for this metric throughout training, demonstrating that its normalized singular values remain closer to uniform and that spectral components are learned at a more balanced rate.

---

> ### Author Response · Authors · 2025-11-21
>
> > W5: “Like the spectral dynamics visualized and derived for linear networks, the paper would benefit from a similar analysis in the experiment section that goes beyond just worst-group/class accuracy with different optimizers.”
>
> Thanks for the suggestion. We have **added an experiment** for the CIFAR-10 dataset with class imbalance in **App. D.10 (Fig. 32)** in the paper, where we go beyond minority and majority class accuracies, and track the singular values of the $k\times n$ logit matrix using $n$ samples across $k/2$ majority and $k/2$ minority classes as training progresses. We follow the same procedure as mentioned in the response to W3 above (see App. D.10 for details), and observe that Muon promotes more balanced learning of spectral components as it maintains a lower KL divergence value throughout training compared to NMD and Signum.
>
> > W4: “In Proposition 2, the initialization for matrices is extremely specific, and it is unclear what all the values mean. This is unlike Proposition 1 and Theorem 1, where the initialization was zero. This makes me question the effect of initialization on the results.”
>
> The spectral initialization used in Proposition 2 is standard in prior works analyzing GD dynamics for deep linear models [1-3].
>
> As the initialization scale of the spectral initialization (where the full model is initialized at $e^{-\delta}UV^\top$) becomes smaller, i.e., for very large $\delta$, this initialization approximates small random initialization.
>
> Regarding your comment about consistency with Prop. 1 and Theorem 1 for linear models where we initialized at zero, it can be easily shown that these results for linear models also hold when we use the spectral initialization that we use for deep linear models. We mainly use zero initialization for linear models because doing so makes the results cleaner to state. In contrast, for models with two or more layers, the gradients for each layer depend on other layers. This can be seen from the gradient expressions in the proofs in App. B.5 and B.6. Therefore, we cannot use exact zero initialization, and hence consider a small-scale initialization.
>
> References:
>
> [1] Gidel, G., Bach, F., and Lacoste-Julien, S. Implicit regularization of discrete gradient dynamics in linear neural networks. NeurIPS 2019.
>
> [2] Saxe, A. M., McClelland, J. L., and Ganguli, S. Exact solutions to the nonlinear dynamics of learning in deep linear neural networks. 2013.
>
> > Q2: Insight into what about the class imbalance/spectra dictates the speedup in generalizing on minority classes with Muon.
>
> We first note that our theoretical analysis presents clean insight into how SpecGD, which is Muon without momentum and with perfect matrix operations, learns the spectral components of the data at the same rate, whereas GD learns more dominant spectral components first. Specifically, the rate of learning for GD is determined by $\tfrac{s^{yx}_i}{s^{xx}_i}$, the ratio of the singular values of the moment matrices $\mathbb{E}[yx^\top]$ and $\mathbb{E}[xx^\top]$, respectively. In the class imbalance setting, with our data model (DM), with class priors $p_1>\dots>p_k$, we show that $s^{yx}_i=\mu p_i$, while $s^{xx}_i=\mu^2 p_i+\sigma^2$. This implies that the singular value ratios $\tfrac{s^{yx}_1}{s^{xx}_1}>\dots>\tfrac{s^{yx}_k}{s^{xx}_k}$ follow the same ordering as the class priors. This means that for GD, the rate of learning different classes directly depends on the class priors (majorities are learned first). In contrast, for SpecGD, learning different spectral components at the same rate translates into learning majority and minority classes at the same rate.
>
> Further, since SpecGD is a canonical form of Muon, these theoretical insights about SpecGD can be translated to Muon. To support this, we have **added experiments** showing that SpecGD and Muon behave similarly in the linear setting, in **App. C (Figs. 10 and 12)** in the paper.

---

> ### Author Response · Authors · 2025-11-21
>
> > Q1: “I am not sure what to read from the eigenvalue and eigenvector figures.”
>
> In Fig. 1, we present the eigenvalues and eigenvectors for the empirical moment matrix $\hat{\mathbb{E}}[xx^\top]$ for the Colored-MNIST dataset, where the digit feature is fully correlated with the label, whereas the color feature is spuriously correlated.
>
> We show that the color feature corresponds to the dominant eigenvalues, while the digit feature information is encoded in less dominant eigenvalues as follows. First, in the eigenvectors plot, the first two columns show a complete separation in the color channels. Second, we consider the reconstructions of two example images from the dataset. We see that reconstructions with the first one or two spectral components contain the color information, but the digit information starts becoming more apparent only as we increase the number of spectral components used for reconstruction.
>
> Additionally, in the leftmost plot in Fig. 1, we see that Muon generalizes on the minority groups, where only the digit feature corresponds to the correct label, faster than other optimizers, which suggests that it learns different spectral components of the data at a more balanced rate.
>
>
> > Q1: “Since the authors are arguing that Muon/Shampoo perform differently than Euclidean methods, can they add Adam to Fig. 1 (left) as well? Also add it to Fig. 3 and others.”
>
> Thanks for the question. In Figs. 1 and 3, we compare Muon (steepest momentum descent with spectral norm) to Signum (steepest momentum descent with $\max$ norm). We compare with Signum instead of Adam here, to examine the effect of the geometry of the norm constraint (spectral vs max), in isolation from the effect of using preconditioner accumulations (as used in Adam). For completeness, we do include comparisons with Adam in these settings in **App. D.1 (Fig. 15)** and **D.2 (Figs. 18 and 19)**, respectively.
>
> We have **updated Section 3** in the paper to clarify the connections between different optimizers.
>
> > Q5: “Why is momentum in Adam insufficient to leverage spectral information in the dataset? Experiments do show that Muon is not always better than Adam, e.g., in Fig. 8 at the end.”
>
> That’s an interesting question. In our work, we don't claim anything specific about the dynamics of Adam/SignGD for learning different spectral components of the data. We include experimental comparisons with Adam for completeness as it is a strong baseline. Regarding your comment about Muon not always outperforming Adam, we note that there are two factors at play here: i) Adam belongs to the SignGD family, whereas Muon belongs to the SpecGD family, and ii) Adam uses accumulation of second moments for preconditioning, whereas Muon uses matrix whitening on the momentum term (no separate accumulation of the preconditioners). We conjecture that an analogous version of Muon that accumulates preconditioner history like Adam could potentially outperform Adam. But this comparison is beyond the scope of this work, as our focus is on comparing the effect of geometry of the norm constraint in steepest descent updates, and in particular SpecGD/Muon vs (N)GD/NMD.
>
> > Q4: “In Proposition 1, what are the implications of the step size value?”
>
> Prop 1 characterizes the discrete-times dynamics of SpecGD iterates. The step size determines the rate of convergence to the final solution, but does not change the qualitative behavior of the dynamics, i.e., SpecGD learns all spectral components at the same rate until saturation, as long as $\eta<\min_i \tfrac{s^{yx}_i}{s^{xx}_i}$, to ensure stable learning.
>
> > Q3: “In Fig. 4, what is the structure of the linear model—just Wx?”
>
> Yes, we have edited Section 3.1 to clarify this.
>
> > T1: “Can the authors briefly mention how Shampoo/Muon reduce to SpecGD and under what assumptions?”
>
> Shampoo and Muon without accumulations and with perfect matrix operations reduce to SpecGD. We have mentioned this in line 53 in the paper and added further clarification on this in **Section 3 and App. A** in the updated version.
>
> Thanks a lot for your time and efforts to review the paper. We appreciate your positive endorsement and support.

---

### Official Review · Reviewer_sqYt · 2025-11-01

**Soundness:** 3
**Presentation:** 4
**Contribution:** 3
**Rating:** 8
**Confidence:** 4

**Summary:**

This paper tries to dive deep into the effectiveness of the optimizers which consider Spectral properties of the parameters like specifically MuOn. The authors study SpecGD which can be seen as an approximation of Muon optimizer. Authors show that the SpecGD method is effective on the imbalanced datasets, as it tends to learn the all the data moments at almost equal rate in comparison to NGD which tends to learn more aggressively from the data moments of the majority classes. They show that this effect is also prominent for deep networks theoretically, and show experimental validations for the same.

**Strengths:**

1.	The paper is well-written and quite comprehensive.
2.	Nice intuitive experimentation is done for each claim in theory which makes understanding easy and also shows real world applicability.
3.	The results of the paper uncover the reasons for effectiveness of the spectral optimizers like MuOn, with faster convergence on imbalanced datasets.

**Weaknesses:**

There is a line of work that shows that the generalization on imbalanced data could be improved by escaping saddle points. Hence, considering algorithms like Perturbed Gradient Descent (PGD) or Sharpness Aware Minimization could be used for optimization and show some connection with.

[R1] Jin, Chi, et al. "How to escape saddle points efficiently." International conference on machine learning. PMLR, 2017.
[R2] Rangwani, Harsh, et al. "Escaping saddle points for effective generalization on class-imbalanced data." Advances in Neural Information Processing Systems 35 (2022): 22791-22805.

**Questions:**

Could the authors explain more about the applicability of the data model they have considered with orthogonal means and specific variance’s generalizability in the real-world datasets?

---

> ### Author Response · Authors · 2025-11-21
>
> Thank you for your positive comments about our work. We are encouraged that you find our results helpful for explaining Muon’s effectiveness in imbalanced settings, applicable to real-world scenarios, and well supported by experiments, and our paper well-written and comprehensive.
>
> > Q1: Could the authors explain more about the applicability of the data model they have considered with orthogonal means and specific variance’s generalizability in the real-world datasets?
>
> Good question! We consider the data model mainly to illustrate a setting where Condition 1 (joint diagonalizability of the two moment matrices $\mathbb{E}[xx^\top]$ and $\mathbb{E}[yx^\top]$) holds. In [1], the authors validate whether a weaker condition is satisfied for some datasets used in practice. Specifically, they check whether there exist orthonormal matrices $U, V$ such that the empirical moment matrices are jointly diagonalizable as $V(S_{xx}+B)V^\top$ and $US_{xy}V^\top$, respectively, where $\||B\||$ is small with respect to $\||S_{xx}\||$. They find that on datasets like MNIST, CIFAR, etc., this ratio is small (see $\Delta_{xy}$ in Table 1 in [1]), so this can be considered as a reasonable condition. For our data model, it can be shown that the empirical moment matrices (using finite samples instead of population), also satisfy this (weaker) condition. In **App. C (Fig. 13)** in the paper, we have added an experiment where we empirically show that GD/SpecGD dynamics predicted by the theory, under Condition 1 where $\||B\||=0$, closely track the dynamics in a setting where only the weaker version holds.
>
> References:
>
> [1] Gidel, G., Bach, F., and Lacoste-Julien, S. Implicit regularization of discrete gradient dynamics in linear neural networks. NeurIPS 2019.
>
> > W1: There is a line of work that shows that the generalization on imbalanced data could be improved by escaping saddle points. Hence, considering algorithms like Perturbed Gradient Descent (PGD) or Sharpness Aware Minimization could be used for optimization and show some connection with.
>
> Thank you for the pointers. We believe that the line of work on methods designed to improve generalization in imbalanced settings is related but slightly tangential to our focus, which is on the (implicit) generalization benefits of spectrum-aware optimizers in such settings. We do study the effect of one such method, namely simple loss re-weighting, in **App. D.8 (Fig. 29)** in the paper. We compared training with NMD using weighted cross entropy loss where samples from minority groups/classes are assigned larger weights vs training with Muon using standard cross entropy loss, on the Colored-MNIST dataset in **Fig. 29**. We believe that doing further comparisons with other methods explicitly designed to improve generalization, such as PGD, SAM, etc., would be an interesting direction for future work.
>
> Thanks a lot for your time and efforts to review the paper. We appreciate your positive endorsement and support.

---

### Comment · Area_Chair_Uam9 · 2025-11-28

Dear Reviewers,

The discussion phase is now underway, and the authors have finished uploading their responses to reviewers. If you haven't already, please carefully review the authors' responses to understand their perspectives. Engage in thoughtful, constructive discussions with authors, sharing your thoughts and seeking clarifications. Please also update your review or rating if necessary.

It is noted in the guideline that reviewers can leave comments visible to authors **until Dec 2 11:59pm AoE**. Your active participation and contribution to the ongoing discussion are highly encouraged. Thank you very much for your contribution to ICLR.

Best regards,

AC

---

### Meta-Review · Area_Chair_Uw5R · 2026-01-06

**Summary:**

This paper theoretically and empirically studies SpecGD on the problem of imbalanced data. It shows that when trained on imbalanced data, the model learns different classes at the same rate. This is a nice result (although a little trivial because what the SpecGD does is exactly to normalize the learning rate of different singular values). Most reviewers support this paper, and one reviewer decided to increase score (though was reverted back). The criticisms from v3RF are good but not so essential in my opinion, and could be addressed in future works.

**Reviewer Concerns:**

I think the reviewer concerns are well addressed, and the remaining ones are not so essential

**Reviewer Scores:**

NA

---

### Decision · Program_Chairs · 2026-01-26

Accept (Poster)